# Distributed Multi-Agent Bandits Over Erdős-Rényi Random Networks

**Jingyuan Liu**[*]
Nanjing University
jingyuanliu@smail.nju.edu.cn

**Hao Qiu**[*]
Università degli Studi di Milano
hao.qiu@unimi.it

**Lin Yang**
Nanjing University
linyang@nju.edu.cn

**Mengfan Xu**[†]
University of Massachusetts Amherst
mengfanxu@umass.edu

## Abstract

We study the distributed multi-agent multi-armed bandit problem with heterogeneous rewards over random communication graphs. Uniquely, at each time step $t$ agents communicate over a time-varying random graph $\mathcal{G}_t$ generated by applying the Erdős–Rényi model to a fixed connected base graph $\mathcal{G}$ (for classical Erdos-Rényi graphs, $\mathcal{G}$ is a complete graph), where each potential edge in $\mathcal{G}$ is randomly and independently present with the link probability $p$. Notably, the resulting random graph is not necessarily connected at each time step. Each agent's arm rewards follow time-invariant distributions, and the reward distribution for the same arm may differ across agents. The goal is to minimize the cumulative expected regret relative to the global mean reward of each arm, defined as the average of that arm's mean rewards across all agents. To this end, we propose a fully distributed algorithm that integrates the arm elimination strategy with the random gossip algorithm. We theoretically show that the regret upper bound is of order $\log T$ and is highly interpretable, where $T$ is the time horizon. It includes the optimal centralized regret $\mathcal{O}\left(\sum_{k:\Delta_k>0} \frac{\log T}{\Delta_k}\right)$ and an additional term $\mathcal{O}\left(\frac{N^2 \log T}{p\lambda_{N-1}(\mathrm{Lap}(\mathcal{G}))} + \frac{KN^2 \log T}{p}\right)$ where $N$ and $K$ denote the total number of agents and arms, respectively. This term reflects the impact of $\mathcal{G}$'s algebraic connectivity $\lambda_{N-1}(\mathrm{Lap}(\mathcal{G}))$ and the link probability $p$, and thus highlights a fundamental trade-off between communication efficiency and regret. As a by-product, we show a nearly optimal regret lower bound. Finally, our numerical experiments not only show the superiority of our algorithm over existing benchmarks, but also validate the theoretical regret scaling with problem complexity.

## 1 Introduction

Multi-armed bandit (MAB) is a widely studied framework for sequential decision-making under uncertainty [Auer et al., 2002]. In this setting, an agent selects an arm from multiple options in each round, observes the reward from the chosen arm, and aims to maximize the cumulative expected reward. The emergence of large-scale cooperative systems holds true in various applications ranging from sensor networks [Ganesan et al., 2004, Zhu et al., 2016] to federated learning [Ye et al., 2023, McMahan et al., 2017] and edge computing [Wang et al., 2022a, Ghoorchian and Maghsudi, 2020]. It has naturally motivated interest in distributed multi-agent multi-armed bandit

---

[*]Equal contribution, in alphabetical order.
[†]Corresponding author

39th Conference on Neural Information Processing Systems (NeurIPS 2025).

(MA-MAB) problem, where multiple agents collaboratively learn to optimize rewards. MA-MAB settings are typically categorized as homogeneous [Landgren et al., 2016a, Martínez-Rubio et al., 2019] or heterogeneous [Zhu et al., 2021, Xu and Klabjan, 2023], depending on whether the reward distributions for the same arm are identical across agents. The heterogeneous setting, in which reward distributions vary across agents, is significantly more general but more challenging, and thus has attracted growing attention. It introduces substantial difficulties, as agents must make sequential decisions under uncertainty while relying on limited information about both their own rewards and the actions or observations of other agents.

Another challenging aspect of MA-MAB lies in the underlying communication protocol, which constrains how agents share information with one another. Decentralized MA-MAB settings are more realistic than centralized ones—where all agents can communicate with any other agent—as they restrict communication to immediate neighbors defined by a graph structure. Much of the existing work has focused on time-invariant graphs [Zhu et al., 2021], where the communication graph remains fixed throughout. However, the complexity of many real-world decentralized systems, such as wireless ad-hoc networks, necessitates the use of time-varying graphs [Zhu and Liu, 2023], particularly random graphs [Xu and Klabjan, 2023]. This added complexity significantly complicates both the communication and learning dynamics. Notably, [Xu and Klabjan, 2023] is the first to consider classical Erdős–Rényi (E-R) random graphs in the MA-MAB setting, where any two agents can communicate with probability $p$ at each time step. However, it is possible that some pairs of agents can never communicate directly due to inherent topological constraints. This scenario has been formulated as a more general version of the E-R graph, where two agents can communicate with probability $p$ only if there is an edge between them in a base graph. Note that when the base graph is a complete graph, it is equivalent to classical (E-R) random graphs, implying consistency. To date, this setting remains unexplored, which motivates our work.

To date, a line of work has studied regret bounds under various graph structures, where connectivity or sequential connectivity is typically required. For example, in the context of time-invariant graphs, Martínez-Rubio et al. [2019] and Zhu et al. [2021] analyze distributed bandits over connected graphs and derive $\log T$ regret bounds. For time-varying graphs, Zhu et al. [2025] obtains $\log T$ regret bounds under the assumption of B-connectivity, where the union of any $l$ consecutive graphs must be connected. In the classical Erdős–Rényi model [Erdos et al., 1960] over fully connected agent communication, Xu and Klabjan [2023] derive a regret of order $\log T$, but only under the assumption that $p \geq 1/2 + 1/2\sqrt{1 - (\epsilon/NT)^{2/N-1}}$ which is larger than $1/2$ and can even approach 1 when the number of agents $N$ or the time horizon $T$ is large. This is a strong assumption, as it may not hold in many real-world settings, but is required in their analysis to ensure that the graph is connected with high probability. Relaxing this connectivity requirement to allow arbitrary $p$ presents a significant challenge. Moreover, their regret bound does not reflect how the link probability $p$ impacts the regret. Incorporating $p$ into the regret expression would significantly improve our understanding of how to choose $p$ in practice—a gap that remains open. In this work, we address both of these gaps. To this end, we address the following key research question: *Can we solve MA-MAB under new Erdős–Rényi random networks and heterogeneous rewards, and derive regret bounds that captures graph complexity under much milder assumptions?*

**Contribution.** We provide an affirmative answer to the above question via the following contributions. Methodologically, we solve the MA-MAB problem over general Erdős–Rényi communication networks with a gossip algorithm, which is widely adopted in distributed settings. Moreover, we adopt an arm elimination based algorithm with a minimal number of arm pulls required for each arm to guarantee sufficient information for each agent, which addresses the E-R communication networks.

Analytically, we study the regret of MA-MAB under our proposed algorithm over general Erdős–Rényi communication networks for any $p \in (0, 1]$, leading to novel contributions. We are the first to 1) explore general Erdos-Rényi graphs induced by any fixed connected base graphs beyond classical setting that assumes a complete base graphs; 2) obtain a tighter and more interpretable regret bound, which generalizes the bound from the homogeneous fixed-graph setting studied in Martínez-Rubio et al. [2019] to our more challenging heterogeneous setting, as shown in Section 5; 3) relax the assumption of a sufficiently large $p$ used in Xu and Klabjan [2023] and reduce the order of $N$ in the upper bound in their analysis of classical Erdős–Rényi models with complete base graphs. Precisely, we obtain a regret upper bound of order of $\mathcal{O}\left(\sum_{k:\Delta_k>0} \frac{\log T}{\Delta_k} + \frac{N^2 \log T}{p\lambda_{N-1}(\text{Lap}(\mathcal{G}))} + \frac{KN^2 \log T}{p}\right)$ where the first term accounts for an optimal centralized regret, aligning with the lower bound established

in Section 5.2, and the last two terms capture the effects of both the link probability $p$ and the algebraic connectivity $\lambda_{N-1}(\mathrm{Lap}(\mathcal{G}))$ of the base graph. Moreover, we uniquely characterize how the regret upper bounds are influenced by $p$ and $\lambda_{N-1}(\mathrm{Lap}(\mathcal{G}))$, which highlights a tradeoff between the communication cost (i.e., the number of communication rounds) and regret performance.

Numerically, we implement our proposed algorithm and conduct experiments to validate our theoretical results with multiple random graph settings and the link probability $p$. We also compare our methods with existing approaches to demonstrate their effectiveness.

## 2 Related Works

**Distributed online algorithm.** Our framework builds on the classical line of work on gossip algorithms [Xiao and Boyd, 2004, Boyd et al., 2006]. Specifically, when an agent has access only to local information and communicates solely with its immediate neighbors, it adopts a gossip algorithm to aggregate information from agents beyond its local neighborhood, based on a weight matrix. Notably, gossip algorithms are widely used in distributed optimization. For example, Duchi et al. [2011], Nedic and Ozdaglar [2009] address distributed convex optimization problems using gossip algorithms. Subsequently, Hosseini et al. [2013], Yan et al. [2013] extend the gossip approach to the online setting and achieve a regret of order $\sqrt{T}$, assuming convex loss functions. Later, Mateos-Núñez and Cortés [2014] consider distributed online optimization over B-connected and design a distributed online primal-dual algorithm coupled with a gossip protocol, also achieving a $\sqrt{T}$ regret bound. More recently, Lei et al. [2020] study the same problem over random communication networks (Erdős–Rényi networks) and obtain the same regret order. They further characterize how regret is affected by the link probability $p$ in the Erdős–Rényi model and the algebraic connectivity of the base graph $\mathcal{G}$. We consider the same graph topology but focus on multi-agent multi-armed bandits, which differ significantly from online convex optimization and introduce the additional challenge of learning the dynamics of bandits. For a more comprehensive survey on distributed online optimization, we recommend the reader to Li et al. [2023] and Yuan et al. [2024].

**Distributed multi-agent multi-armed bandit.** Along the line of work on MA-MAB, several studies [Landgren et al., 2016a,b, Zhu et al., 2020, Chawla et al., 2020, Wang et al., 2022b, 2020, Zhu et al., 2025, Martínez-Rubio et al., 2019, Agarwal et al., 2022, Sankararaman et al., 2019, Zhu et al., 2021, Zhu and Liu, 2023, Xu and Klabjan, 2023, Yi and Vojnovic, 2023] have investigated both homogeneous and heterogeneous settings. In homogeneous settings, numerous works incorporate gossip algorithms to reduce regret in terms of the number of agents; information sharing among agents accelerates the concentration of reward observations. For example, Landgren et al. [2016a,b] firstly formulate this problem and solve it using gossip algorithms. Martínez-Rubio et al. [2019] achieves the optimal centralized regret—independent of the number of agents and matching that of the single-agent bandit—plus an additional term depending on the spectral gap of the communication matrix. Chawla et al. [2020] characterizes the regret-communication tradeoff, considers circular ring graphs, and improves regret. Wang et al. [2022b, 2020] further focus on optimizing communication efficiency to minimize the number of communication rounds while guaranteeing regret performance. In contrast, we consider more challenging heterogeneous settings and also characterize the regret-communication trade-off. Here gossiping enables regret reduction in terms of the order of $T$; without information from other agents, the regret can easily be linear in $T$ [Xu and Klabjan, 2025]. In this direction, Zhu et al. [2021] is the first to study heterogeneous rewards over a time-invariant connected graph and establishes regret bounds of order $\log T$. Zhu and Liu [2023] extend this to B-connected graphs, also achieving regret bounds of order $\log T$. Recently, Xu and Klabjan [2023] propose a gossip-based algorithm for the classical E-R model and obtain regret bounds of order $\log T$ when $p$ is sufficiently larger than $1/2$ to ensure the graph is connected with high probability. More generally, Yi and Vojnovic [2023] consider MA-MAB with heterogeneous rewards in the adversarial environment and establishes a regret bound of order $T^{2/3}$. In contrast, we consider the stochastic setting with general Erdős–Rényi random networks, without any assumption on $p$.

## 3 Setting and Notations

In this section, we formally define the problem of interest, starting with general notations. A full notation chart can be found in Appendix E.

**General Notations.** For a matrix $M \in \mathbb{R}^{p \times q}$, let $[M]_{i,j}$ denote the entry in the $i$-th row and $j$-th column. Given a doubly stochastic matrix $P \in \mathbb{R}^{d \times d}$, we denote by $\lambda_2(P)$ its second largest eigenvalue. Let $e_i \in \mathbb{R}^d$ be the $i$-th standard basis vector, $\mathbf{1} \in \mathbb{R}^d$ the all-ones vector, and $I_d$ the $d \times d$ identity matrix. We use $\mathbb{I}\{\cdot\}$ to denote the indicator function, which equals 1 if the condition inside holds, and 0 otherwise. Moreover, we use $[n] = \{1, 2, \cdots, n\}$ to denote a set of indices.

**Multi-Agent Multi-armed Bandit.** We consider a multi-agent multi-armed bandit (MA-MAB) setting involving $N$ agents. The bandit problem is run over a time horizon of $T$. In each round $t \in [T]$, every agent $i \in [N]$ selects an arm $A_i(t) \in [K]$ and receives a local stochastic reward $X_{i,A_i(t)}(t)$ drawn independently from an unknown, time-invariant distribution $\mathbb{P}_{i,A_i(t)}$ supported on $[0, 1]$, with mean $\mu_{i,A_i(t)} = \mathbb{E}[X \sim \mathbb{P}_{i,A_i(t)}] \in [0, 1]$. However, the agent's true objective depends on the global reward $X_{A_i(t)}(t)$, where $X_{A_i(t)} := \frac{1}{N} \sum_{j \in [N]} X_{j,A_i(t)}(t)$, which is the average reward over all agents and is not observable by any agent. Accordingly, we define the global mean reward for arm $k$ as $\mu_k := \frac{1}{N} \sum_{j \in [N]} \mu_{j,k}$ by taking the expectation of $X_k(t)$, where $\mu_{j,k}$ is the mean of $\mathbb{P}_{j,k}$. The underlying target is to optimize the global mean reward, and hence agents require estimates of the global mean reward values for each arm in order to identify the global optimal arm with the highest global mean reward. We further use $T_{i,k}(t)$ to denote the total number of times agent $i$ has selected arm $k$ up to time $t$ by $T_{i,k}(t) = \sum_{s=1}^{t} \mathbb{I}(A_i(s) = k)$.

Throughout, we focus on a decentralized setting where $N$ agents are distributed on undirected time-varying graphs and communicate via the graphs. Specifically, we consider a communication graph based on *Erdős–Rényi random graphs*. More precisely, agents communicate via a time-varying graph $\mathcal{G}_t = (\mathcal{V}, \mathcal{E}_t)$, where the vertex set $\mathcal{V} = [N]$ is fixed, but the edge set $\mathcal{E}_t$ may vary at each round $t$. Uniquely, each *communication graph* $\mathcal{G}_t$ is generated based on an underlying undirected, fixed, and connected (but not necessarily complete) *base graph* $\mathcal{G} = (\mathcal{V}, \mathcal{E})$ that defines all feasible communication edges. The connectedness of the base graph is essential to avoid linear regret as proved in Theorem 4 in Xu and Klabjan [2025]. In every round $t$, each edge in graph $\mathcal{G}$ is independently in $\mathcal{G}_t$ with probability $p \in (0, 1]$. Two agents can communicate if and only if there is an edge between them in $\mathcal{G}_t$—namely, an active edge. Formally, the random graph generation reads as follows.

**Assumption 3.1** (Erdős–Rényi Random Graph). In each round $t$, the random communication graph $\mathcal{G}_t = (\mathcal{V}, \mathcal{E}_t)$, generated from the base graph $\mathcal{G}$ meets:

$$\mathbb{P}\left((i, j) \in \mathcal{E}_t\right) = \begin{cases} p, & \text{if } (i, j) \in \mathcal{E}, \\ 0, & \text{otherwise}, \end{cases}$$

for all vertices $i, j \in \mathcal{V}$. We refer to $p$ as the *link probability*.

Thus, $\mathcal{E}_t \subseteq \mathcal{E}$ for all $t$. Let $\mathcal{N}_i = \{j \in \mathcal{V} : (i, j) \in \mathcal{E}\}$ denote the neighbors of agent $i$ in the base graph $\mathcal{G}$, and $\mathcal{N}_i(t) = \{j \in \mathcal{V} : (i, j) \in \mathcal{E}_t\}$ denote the active neighbors of agent $i$ in round $t$, clearly satisfying $\mathcal{N}_i(t) \subseteq \mathcal{N}_i$.

The shared objective for each agent is to design a distributed algorithm $\pi$ that minimizes the global regret over $T$ rounds; in other words, all agents share a common target that makes consensus possible and necessary. Precisely, we first define the individual regret, which represents the objective of any agent $i \in [N]$, as:

$$\text{Reg}_{i,T}(\pi) = T\mu^* - \sum_{t=1}^{T} \mu_{A_i(t)} = \sum_{t=1}^{T} \Delta_{A_i(t)}, \tag{1}$$

where $\mu^* = \max_{k \in [K]} \mu_k$ is the optimal global mean reward, and $\Delta_k = \mu^* - \mu_k$ is the global suboptimality gap for arm $k$. Notably, agents aim to optimize the global mean rewards of the pulled arms, i.e., to pull the global optimal arm. We define the individual regret as the cumulative difference between the global mean reward of the global optimal arm and that of the actual arm pulled by each agent. Building on this, we define the global regret of the entire system for algorithm $\pi$ as the sum of the individual regret over all agents:

$$\text{Reg}_T(\pi) = \sum_{i \in [N]} \text{Reg}_{i,T}(\pi) = NT\mu^* - \sum_{i \in [N]} \sum_{t=1}^{T} \mu_{A_i(t)} = \sum_{i \in [N]} \sum_{t=1}^{T} \Delta_{A_i(t)}, \tag{2}$$

which establishes the equivalence of Equation (1) in the context of individual agents.

# 4 Algorithm: Gossip Successive Elimination

In this section, we present our proposed methodology. Unlike existing work on gossip bandits [Martínez-Rubio et al., 2019, Zhu et al., 2021], which assumes fixed connected graphs, we handle the fundamental challenges of randomness and possible disconnectivity in the random communication graph. To this end, we propose distributed bandit algorithm designed for Erdős–Rényi random graphs (Assumption 3.1) based on arm elimination, named *Gossip Successive Elimination (GSE)* (Algorithm 1). GSE combines three key components: an **arm elimination protocol**, a **refined weight matrix**, and a **novel confidence interval design**. Specifically, we adopt the arm elimination framework [Even-Dar et al., 2006] in distributed bandits setting, which naturally ensures that all agents pull each arm a comparable number of times, so that the global estimates obtained via gossip are not largely biased and not dominated by any single agent's feedback. Second, we design a refined weight matrix based on the Laplacian, which guarantees that the weight matrices are i.i.d. across rounds. This construction is crucial to balance information among agents over time and to ensure that the global reward estimates converge to the true global mean reward for each arm. Finally, GSE introduces a new confidence interval with two terms: the first term captures the estimation error due to finite sampling, while the second term accounts for the consensus error, reflecting the approximation introduced by time-delayed information propagation under random gossip. Together, all these novel components enable GSE to jointly address reward heterogeneity and random communication in a principled and analytical paradigm.

The agent needs several parameters, including the link probability $p$ and algebraic connectivity $\lambda_{N-1}(\text{Lap}(\mathcal{G}))$ as inputs. Initially, agent $i$'s active arm set $\mathcal{S}_i$ is set to the full set of arms $[K]$, while both the local reward estimate $\widehat{\mu}_{i,k}(t)$ and global reward estimate $z_{i,k}(t+1)$ are initialized to zero ($t = 0$). Following the standard practice in existing work [Martínez-Rubio et al., 2019, Zhu et al., 2021, Xu and Klabjan, 2023], we assume that $\lambda_{N-1}(\text{Lap}(\mathcal{G}))$ is known, while allowing the link probability $p$ to remain unknown. In this case, the algorithm can still be implemented, and the corresponding regret upper bound preserved, by adding a short burn-in phase to estimate a value $\hat{p} \in (p/2, p]$. This estimation requires only $\mathcal{O}(\log T)$ time steps. We refer the reader to Appendix C for the details and theoretical analysis of this procedure.

At each round $t$, agent $i$ observes its active neighbors $\mathcal{N}_i(t)$ in the communication graph $\mathcal{G}_t$, which is randomly generated from the base graph $\mathcal{G}$ according to Assumption 3.1. As part of our key contributions, we consider a weighting matrix $W_t$ for gossip as

$$W_t = I_N - \frac{1}{N}\text{Lap}(\mathcal{G}_t), \tag{3}$$

where $\text{Lap}(\mathcal{G}_t)$ denotes the Laplacian matrix of the communication graph $\mathcal{G}_t$.

---

**Algorithm 1** Gossip Successive Elimination for Agent $i \in [N]$

---

1: **Input:** Algebraic connectivity $\lambda_{N-1}(\text{Lap}(\mathcal{G}))$, total time horizon $T$, set of arms $[K]$, link probability $p$
2: **Initialization:** Active set $\mathcal{S}_i \leftarrow [K]$, local reward estimate $\widehat{\mu}_{i,k}(0) \leftarrow 0$, global reward estimate $z_{i,k}(1) \leftarrow 0$ for all $k \in [K]$.
3: **for** $t = 1, 2, \ldots, T$ **do**
4:     Select arm $A_i(t) \in \mathcal{S}_i$ with the minimum pull count $T_{i,k}(t)$ and update $T_{i,k}(t)$
5:     Receive feedback and update statistics for each arm using Equation (4)
6:     Remove arm $k \in \mathcal{S}_i$ if there exists an arm $k' \in \mathcal{S}_i$, $k' \neq k$, satisfying the elimination condition in Equation (7)
7:     Update the active set $\mathcal{S}_i$ according to Equation (8)
8: **end for**

---

The execution steps of agents run as follow. Agent $i$ selects arm $A_i(t)$ from the active set $\mathcal{S}_i$ with the least number of pulls $T_{i,k}(t)$, observes the local reward of $A_i(t)$, and updates the reward estimates as follows:

$$\widehat{\mu}_{i,k}(t) = \frac{1}{T_{i,k}(t) \vee 1} \sum_{\tau=1}^{t} \mathbb{I}\{A_i(\tau) = k\} X_{i,k}(\tau),$$

$$z_{i,k}(t+1) = \sum_{j \in \mathcal{N}_i(t) \cup \{i\}} [W_t]_{i,j} z_{j,k}(t) + \widehat{\mu}_{i,k}(t) - \widehat{\mu}_{i,k}(t-1), \tag{4}$$

where $X_{i,k}(\tau)$ is the feedback observed by agent $i$ from pulling arm $k$ at round $\tau$. The global estimate $z_{i,k}(t)$ is updated via a gossip protocol: at each round $t$, agent $i$ aggregates its own and its active neighbors' estimates, weighting each neighbor $j \in \mathcal{N}_i(t) \cup \{i\}$ by the corresponding entry $[W_t]_{i,j}$ of the matrix $W_t$. We also define the gossip-based upper and lower confidence bounds for arm $k$, which remain key criteria for arm elimination and updating $\mathcal{S}_i$, as

$$\mathrm{GUCB}_{i,k}(t) = z_{i,k}(t) + c_{i,k}(t), \ \mathrm{GLCB}_{i,k}(t) = z_{i,k}(t) - c_{i,k}(t).$$

Here the confidence bound $c_{i,k}(t)$ reads as

$$c_{i,k}(t) = \sqrt{\frac{4\log(T)}{N \max\{T_{i,k}(t) - KL^*, 1\}}} + \frac{4(\sqrt{N} + \tau^*)}{\max\{T_{i,k}(t) - KL^*, 1\}}, \tag{5}$$

with

$$\tau^* = \left\lceil \frac{2N\log(T)}{p\lambda_{N-1}(\mathrm{Lap}(\mathcal{G}))} \right\rceil, \ L^* = N \left\lceil -\frac{2\log(NT)}{\log(1-p)} \right\rceil. \tag{6}$$

The confidence radius in Equation (5) decomposes into two parts. The first term captures the **estimation error**, arising from the statistical variance of independently sampled rewards for each arm. The second term captures the **consensus error**, which stems from the cumulative approximation error error incurred as agents reach consensus via a gossip-based communication protocol.

Arm elimination occurs if and only if there exists an arm $k' \neq k$ in $\mathcal{S}_i$ that meets the condition

$$\mathrm{GLCB}_{i,k'}(t) \geq \mathrm{GUCB}_{i,k}(t), \tag{7}$$

which means that arm $k'$ has a higher global reward estimate than arm $k$ with high probability. Subsequently, the active set $\mathcal{S}_i$ is updated as

$$\mathcal{S}_i \leftarrow \bigcap_{j \in \mathcal{N}_i(t) \cup \{i\}} \mathcal{S}_j. \tag{8}$$

This procedure coupled with the arm-elimination strategy ensures that, for every agent, each arm in the active set is pulled a roughly equal number of times, ensuring consensus.

## 5 Regret Analyses

In this section, we analyze the regret of the proposed algorithm and establish an upper bound on the corresponding regret, demonstrating its theoretical effectiveness. Additionally, we derive a lower bound for our problem setting, highlighting the problem's inherent complexity and showing that the algorithm is nearly optimal up to some interpretable factors.

### 5.1 Upper Bound

We start by presenting the regret upper bound for Algorithm 1. To that end, we first introduce several technical lemmas that play a key role in the regret analysis. The proofs can be found in Appendix B.

**Lemma 5.1.** *Let us assume that the communication graph follows Assumption 3.1. Then we have that for Algorithm 1, for any agent $i \in [N]$, any arm $k \in [K]$, and any $t \in [T]$, the following holds with probability at least $1 - \frac{3NK}{T}$,*

$$|z_{i,k}(t) - \mu_k| \leq c_{i,k}(t),$$

*where $c_{i,k}(t)$ is the confidence bound defined in Equation (5), and $\tau^*$ and $L^*$ are the parameters introduced in Equation (6).*

Importantly, we show that the estimation error compared to the global mean value is upper bounded by the confidence bound $c_{i,k}(t)$. Notably, $c_{i,k}(t)$ is monotonically decreasing in the number of pulls of arm $k$ by agent $i$, i.e., $T_{i,k}(t)$, when $T_{i,k}(t) > KL^*$. Here, $KL^*$ is the minimal number of pulls required to ensure that agent $i$ has collected sufficient information from all other agents. This also implies that the global estimate $z_{i,k}(t)$ becomes increasingly accurate and approaches the true mean reward $\mu_k$ as the number of pulls increases. As a result, based on Lemma 5.1, we derive the following regret bound for individual agents.

**Lemma 5.2.** *Let us assume that the communication graph follows Assumption 3.1. The individual regret of agent $i$ defined in Equation (1) for Algorithm 1 is bounded as:*

$$\text{Reg}_{i,T}(\text{GSE}) \leq \sum_{k:\Delta_k>0} \left( \frac{64 \log(T)}{N\Delta_k} + 16\left(\sqrt{N} + \tau^*\right) \right) + KL^* + 3KN\Delta_{\max},$$

*where $\Delta_{\max} = \max_{k\in[K]} \Delta_k$ denotes the largest reward gap across all arms.*

In other words, Lemma 5.2 shows that the regret incurred by any individual agent can be effectively bounded. This result naturally extends to the global regret, as stated in the theorem below.

**Theorem 5.3.** *Let us assume that the communication graph follows Assumption 3.1. The global regret defined in Equation (2) for Algorithm 1 is bounded as:*

$$\text{Reg}_T(\text{GSE}) = \sum_{i\in[N]} \text{Reg}_{i,T}(\text{GSE}) \leq \mathcal{O}\left( \sum_{k:\Delta_k>0} \frac{\log(T)}{\Delta_k} + \frac{N^2 \log(T)}{p\lambda_{N-1}(\text{Lap}(\mathcal{G}))} + \frac{KN^2 \log(NT)}{p} \right),$$

*where $\lambda_{N-1}(\text{Lap}(\mathcal{G}))$ is the second smallest eigenvalue of $\text{Lap}(\mathcal{G})$.*

We continue our discussion on how the base graph $\mathcal{G}$ topology affects the regret bound. In addition to the parameter $p$, which determines the difference between $\mathcal{G}$ and $\mathcal{G}_t$, another key factor is $\lambda_{N-1}(\text{Lap}(\mathcal{G}))$, which reflects the topology of the base graph $\mathcal{G}$. This value is the algebraic connectivity or Fiedler value of $\mathcal{G}$, which reflects how well connected the overall graph is. To illustrate this, we next provide more explicit regret upper bounds by specifying $\lambda_{N-1}(\text{Lap}(\mathcal{G}))$ for different base graph topologies commonly used in distributed optimization Duchi et al. [2011]. The following corollary summarises how the choice of the random gossip matrix in Equation (3) interacts with different topologies of the base graph. The proof of $\lambda_{N-1}(\text{Lap}(\mathcal{G}))$ for various base graphs $\mathcal{G}$ can be found in Corollary 1 of Duchi et al. [2011].

**Corollary 5.4.** *For specific choices of the base graph $\mathcal{G}$, the global regret upper bound in Theorem 5.3 simplifies as follows:*

*1) When $\mathcal{G}$ is a complete graph with $\lambda_{N-1}(\text{Lap}(\mathcal{G})) = N$, and refining $L^* = \left\lceil -\frac{2\log(NT)}{\log(1-p)} \right\rceil$ in Equation (6), Theorem 5.3 simplifies to: $\mathcal{O}\left( \sum_{k:\Delta_k>0} \frac{\log T}{\Delta_k} + \frac{KN \log T}{p} \right)$.*

*2) When $\mathcal{G}$ is a $\sqrt{N} \times \sqrt{N}$ 2D grid with $\lambda_{N-1}(\text{Lap}(\mathcal{G})) = 2\left(1 - \cos\left(\frac{\pi}{\sqrt{N}}\right)\right) = \Theta(1/N)$, Theorem 5.3 simplifies to: $\mathcal{O}\left( \sum_{k:\Delta_k>0} \frac{\log T}{\Delta_k} + \frac{N^2(K+N)\log T}{p} \right)$.*

*3) When $\mathcal{G}$ is an expander graph with a bounded ratio between minimum and maximum node degrees and thus $\lambda_{N-1}(\text{Lap}(\mathcal{G})) = \Theta(1)$, Theorem 5.3 simplifies to: $\mathcal{O}\left( \sum_{k:\Delta_k>0} \frac{\log T}{\Delta_k} + \frac{KN^2 \log T}{p} \right)$.*

*Remark* 5.5 (Comparison of Regret Bounds). In Theorem 5.3, for any fixed connected base graph $\mathcal{G}$ and $0 < p \leq 1$, we obtain the optimal centralized regret $\mathcal{O}\left( \sum_{k:\Delta_k>0} \frac{\log T}{\Delta_k} \right)$ (see the lower bound in Section 5.2) plus an additional term $\mathcal{O}\left( \frac{N^2 \log T}{p\lambda_{N-1}(\text{Lap}(\mathcal{G}))} + \frac{KN^2 \log T}{p} \right)$. We emphasize that our regret bound outperforms existing work—many of which are special (degenerated) cases of our more general framework—and is easier to interpret. Notably, Xu and Klabjan [2023] study MA-MAB under the classical E-R model (where $\mathcal{G}$ is a complete graph) with $p$ largely over $1/2$, and derive a regret bound of $\mathcal{O}\left( \sum_{k:\Delta_k>0} \frac{N \log T}{\Delta_k} \right)$. In the same setting, based on Corollary 5.4, we obtain a regret bound of $\mathcal{O}\left( \sum_{k:\Delta_k>0} \frac{\log(T)}{\Delta_k} + KN \log(T) \right)$, which is significantly smaller. Also, Zhu and Liu [2023] consider B-connected graphs, which do not capture E-R random graphs; the distinction between the two models is discussed in detail in Yuan et al. [2024]. Finally, when $p = 1$, the communication graph becomes time-invariant, which corresponds to most existing work where connected graphs are assumed. For example, Zhu et al. [2021] and Zhu and Liu [2023] study such settings. The former derive a regret bound of $\mathcal{O}\left( \sum_{k:\Delta_k>0} \frac{N^2 \log T}{\Delta_k} \right)$, which is worse than our results with a dependency on $N$. The latter obtain $\mathcal{O}\left( \max\left( \sum_{k:\Delta_k>0} \frac{N \log T}{N_k \Delta_k}, K_1, K_2 \right) \right)$, where $K_1$ and $K_2$ depend on $T$ but lack explicit formulas, may grow arbitrarily large, and are difficult to interpret—at least to the best of our knowledge.

*Remark* 5.6 (Regret and Communication Trade-off). We emphasize that our regret upper bound exhibits a novel trade-off between regret performance and communication efficiency in the presence of random graphs. It is straightforward to observe that increasing $p$ reduces the regret bound but increases communication overhead, thereby lowering communication efficiency. For classical E-R graphs, the expected number of agent-to-agent communications per agent over the given time horizon is $pNT$, while the expected regrets decrease as $p$ increases. Therefore, for a fixed time horizon $T$ and a given base graph $\mathcal{G}$, $p$ can be tuned to balance communication overhead and reward maximization, informing practical decision making.

## 5.2 Lower Bound

In this section, we establish a lower bound on the global regret, as defined in Equation (2). Unlike the upper bound, this lower bound applies to any reasonable algorithm under a specific problem instance, highlighting the problem complexity. Detailed proofs are deferred to Appendix B. We begin by introducing several definitions related to the problem instance and the notion of a reasonable algorithm.

**Definition 5.7** (Gaussian Instance). An instance $\nu$ is called a *Gaussian instance* if, for every agent $i \in [N]$ and arm $k \in [K]$, the reward distribution $\mathbb{P}_{i,k}$ is Gaussian with unit variance.

**Definition 5.8** (Consistent Policy). Let $\mathcal{I}$ denote a class of problem instances, and let $\mathrm{Reg}_T^\nu(\pi)$ denote the regret incurred by policy $\pi$ on instance $\nu$. A policy (algorithm) $\pi$ is said to be *consistent* on $\mathcal{I}$ if there exist constants $C > 0$ and $s \in (0, 1)$ such that $\mathrm{Reg}_T^\nu(\pi)$ meets $\mathrm{Reg}_T^\nu(\pi) \leq CT^s$ for all instances $\nu \in \mathcal{I}$.

We next present the problem instance constructed to establish the regret lower bound. Note that an alternative, equivalent expression of the global regret reads as

$$\mathrm{Reg}_T^\nu(\pi) = \sum_{k \in [K]} \Delta_k \sum_{i \in [N]} \mathbb{E}[T_{i,k}(T)].$$

Thus, to derive a lower bound on regret, it suffices to lower bound the total expected number of pulls $\sum_{i \in [N]} \mathbb{E}[T_{i,k}(T)]$ for suboptimal arms $k \in [K]$. To this end, we consider a Gaussian instance $\nu$ where each $\mathbb{P}_{i,k} = \mathcal{N}(\mu_{i,k}, 1)$ with the random graph communication protocal based any connected base graph $\mathcal{G}$ and connection probability $p$, and construct a perturbed instance $\nu'$ such that:

$$\mathbb{P}'_{i,a} = \begin{cases} \mathcal{N}(\mu_{i,a}, 1), & \text{if } a \neq k, \\ \mathcal{N}(\mu_{i,a} + (1+\varepsilon)\Delta_a, 1), & \text{if } a = k, \end{cases}$$

for a small constant $\varepsilon \in (0, 1)$ representing the level of perturbation. The communication protocal for $\nu'$ is the same as that for $\nu$. Under this perturbation, which defines a new problem instance, we derive the following information-theoretic inequality:

$$\sum_{j \in [N]} \mathbb{E}[T_{j,k}(T)] \cdot \frac{(1+\varepsilon)^2 \Delta_k^2}{2} \geq \log\left(\frac{NT\varepsilon\Delta_k}{4\left(\mathrm{Reg}_T^\nu(\pi) + \mathrm{Reg}_T^{\nu'}(\pi)\right)}\right), \tag{9}$$

which imposes a lower bound on the total number of pulls of arm $k$ across all agents.

By applying this inequality to a consistent policy $\pi$ and rearranging the terms, we obtain the following lower bound on regret. The formal statement reads as follows.

**Theorem 5.9.** *Let $\pi$ be a consistent policy on the class $\mathcal{I}$ of Gaussian instances for some $s \in (0, 1)$. Then, for all instances $\nu \in \mathcal{I}$ and any $\varepsilon \in (0, 1]$, the following holds:*

$$\lim_{T \to \infty} \frac{\mathrm{Reg}_T^\nu(\pi)}{\log T} \geq \sum_{k:\Delta_k > 0} \frac{2(1-s)}{(1+\varepsilon)^2 \Delta_k}.$$

*Remark* 5.10 (Comparison with Upper Bounds). Recall that the regret upper bound in Theorem 5.3 consists of two terms: a centralized term and additional terms that capture the influence of the communication graph. The lower bound in Theorem 5.9 shows that the centralized component, $\mathcal{O}\left(\sum_{k:\Delta_k > 0} \frac{\log T}{\Delta_k}\right)$, is tight, thereby establishing the optimality of the centralized regret achieved by Algorithm 1. This result is intuitive: the problem effectively involves $N$ agents collaboratively solving

a global multi-armed bandit task. With appropriate information sharing, the collective performance can match that of a single-agent bandit problem with full access to all rewards. Hence, achieving a global regret of the same order as the classical centralized bandit setting is both natural and optimal. The additional term related to the communication graph, though not captured in Theorem 5.9, can be interpreted intuitively. Consider, for instance, a circular base graph $\mathcal{G}$: on average, it takes $O(N/p)$ rounds for information from one agent to propagate to all others. Aggregating over all $N$ agents, this leads to an unavoidable lower bound of $O(N^2/p)$ in the global regret. Providing a formal lower bound proof that fully characterizes this effect remains an interesting direction for future work.

## 6  Experiments

In this section, we demonstrate the effectiveness of our algorithm through numerical experiments on both synthetic and real-world datasets.[3] The objective is twofold. First, we show that the cumulative regret of our algorithm grows logarithmically with respect to $T$ and is significantly smaller than that of existing benchmarks, thereby validating our theoretical findings. We use DrFed-UCB, proposed by [Xu and Klabjan, 2023], as the baseline. Second, we conduct a simulation study to examine how the regret depends on the link probability $p$ and the algebraic connectivity of the base graph $\mathcal{G}$, as reflected in the regret bound. We evaluate the impact of different values of $p$ across various base graphs, including the complete graph, grid, and Petersen graph [Holton and Sheehan, 1993]. Each edge in the base graph $\mathcal{G}$ appears in the communication graph $\mathcal{G}_t$ with probability $p$. Moreover, we provide additional experiments (i.e., an ablation study) in Appendix D, illustrating the regret dependency on the key parameters $p$ and $\lambda_{N-1}(\text{Lap}(\mathcal{G}))$ that determine the problem complexity beyond $T$, to validate our theoretical findings.

**Experimental Settings.**  For synthetic experiment setting, We set $T = 10000$, $N = 16$, and $K = 5$; for the Petersen graph, we use $N = 10$ by definition. For the comparison with DrFed-UCB, we consider a complete graph and a high link probability ($p = 0.9$), as required therein. Before the game starts, we sample each $q_i$ independently and uniformly from the interval $[0, 1]$ for each agent $i$. The local mean reward of arm $k$ on agent $i$ is given by $\mu_{i,k} = q_i \cdot \frac{k-1}{K-1}$, and the global mean reward of arm $k$ is $\mu_k = \frac{k-1}{K-1} \cdot \frac{\sum_{i \in [N]} q_i}{N}$. At each time step $t$, each agent $i \in [N]$ selects an arm and observes the local reward. For real-world experiments, we use the MovieLens dataset and refer to Yi and Vojnovic [2023] for details. We set the horizon $T = 10,000$, and select 20 users as agents ($N = 20$) and 5 genres as arms ($K = 5$). At each time step $t$, each agent randomly selects a movie from the genres. All ratings (rewards) of movies are normalised to $[0, 1]$.

**Experimental Results.**  All experiments are performed with 20 independent replications. The shaded areas consider a range centered around the mean with half-width corresponding to the empirical standard deviation over 20 repetitions. In Figures 1a and 1b, we observe that our algorithm consistently outperforms DrFed-UCB on both synthetic and real-world datasets. In all runs, after an initial exploration period, our algorithm eliminates a significant number of suboptimal actions, resulting in near-constant regret thereafter. In Figures 1c and 1d, we observe that increasing the link probability $p$ improves the algorithm's performance, clearly validating the regret–communication trade-off. Additionally, different base graphs significantly impact the regret under the same $p$ value—with the complete graph yielding the lowest regret.

## 7  Conclusion and Future Work

In this paper, we study the multi-agent multi-armed bandit (MA-MAB) problem under general Erdős–Rényi random networks with heterogeneous rewards. To the best of our knowledge, we are the first to formulate MA-MAB with Erdős–Rényi random networks, where the communication graph is induced by a base graph and each edge in the base graph appears in the communication graph with probability $p$. This formulation generalizes the classical Erdős–Rényi model, in which the base graph is complete. We propose an algorithmic framework that incorporates the gossip communication protocol into arm elimination. Importantly, we analyze the regret bound of the algorithm and show that it improves the regret even under the classical Erdős–Rényi model. Moreover, our regret bound

---

[3]The code for the experiments is available at https://github.com/haoqiu95/multi-agent-bandit.

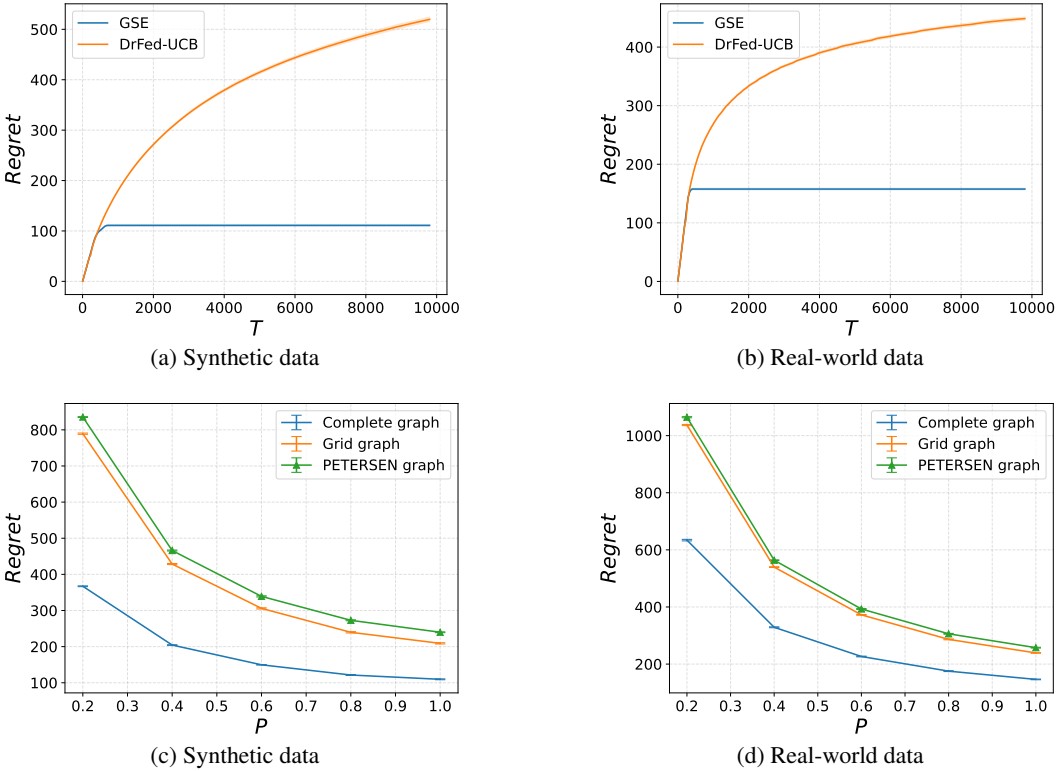

Figure 1: Top two: comparison of the empirical results of our algorithm and DrFed-UCB (complete base graph, link probability $p = 0.9$). Bottom two: regret of our algorithm on different base graphs as $p$ varies.

holds for any $p$, explicitly characterizes its dependency on $p$ and the algebraic connectivity of the base graph. This naturally reveals a trade-off between regret and communication efficiency. Moving forward, while we focus extensively on the stochastic setting, it would be valuable and exciting to explore other reward models—such as contextual bandits, where rewards depend on dynamic, non-stationary contexts. In addition, achieving the optimal trade-off among regret, communication, and privacy, as previously studied in homogeneous MA-MAB settings, points out a meaningful direction for future research.

## Acknowledgments and Disclosure of Funding

Hao Qiu acknowledges the financial support from the EU Horizon CL4-2022-HUMAN-02 research and innovation action under grant agreement 101120237, project ELIAS (European Lighthouse of AI for Sustainability) and from the One Health Action Hub, University Task Force for the resilience of territorial ecosystems, funded by Università degli Studi di Milano (PSR 2021-GSA-Linea 6).

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

# A  Auxiliary results

In this section, we show several auxiliary lemmas that will be helpful throughout the paper.

The following lemma is the concentration bound for the estimation of the observed expected rewards,

**Lemma A.1.** *Let $\widehat{\mu}_{i,k}(t)$ be the observed empirical average of the expected reward up to the end of round $t-1$. Then,*

$$\mathbb{P}\left[|\widehat{\mu}_{i,k}(t) - \mu_{i,k}| > \sqrt{\frac{2\log T}{T_{i,k}(t)}}\right] \leq \frac{2}{T^2}$$

*for $i \in [N]$, $k \in [K]$ and $t \in [T]$ holds.*

*Proof of Lemma A.1.* This lemma follows immediately from Hoeffding's inequality and a union bound. □

The following lemma is the concentration bound for the estimation of the observed expected rewards.

**Lemma A.2.** *Let $\widehat{\mu}_{i,k}(t)$ be the observed empirical average of the expected reward for agent $i$ pulling arm $k$. Then,*

$$\mathbb{P}\left[\left|\sum_{i\in[N]}\widehat{\mu}_{i,k} - \sum_{i\in[N]}\mu_{i,k}\right| > \sqrt{\frac{4N\log T}{\min_{i\in[N]}\{T_{i,k}(t)\}}}\right] \leq \frac{2}{T^2}$$

*holds for any $i \in [N], k \in [K]$ and $t \in [T]$.*

*Proof of Lemma A.2.* We sample $T_{i,k}(t)$ number of random variables iid from $\mathbb{P}_{i,k}$ for each $i \in [N]$, each taking values in $[0,1]$, and hence $X_{i,k}(\tau) - \mu_{i,k}$ is 1-subgaussian for each $\tau$.

According to the property of subgaussian variables, we can obtain that

$$\sum_{i\in[N]}(\widehat{\mu}_{i,k}(t) - \mu_{i,k}) \text{ is } \left(\sum_{i\in[N]}\frac{1}{T_{i,k}(t)}\right)^{1/2} \text{ - subgaussian.}$$

Moreover, note that $\sum_{i\in[N]}\frac{1}{T_{i,k}(t)} \leq \frac{N}{\min_{i\in[N]}T_{i,k}(t)}$, then by applying Chernoff's bound, we have

$$\mathbb{P}\left(\left|\sum_{i\in[N]}\widehat{\mu}_{i,k}(t) - \mu_{i,k}\right| \geq \epsilon\right) \leq 2\exp\left(-\frac{\epsilon^2 \min_{i\in[N]}\{T_{i,k}(t)\}}{2N}\right).$$

Taking $\epsilon = \sqrt{\frac{4N\log T}{\min_{i\in[N]}\{T_{i,k}(t)\}}}$, we obtain that

$$\mathbb{P}\left(\left|\sum_{i\in[N]}\widehat{\mu}_{i,k}(t) - \mu_{i,k}\right| \geq \sqrt{\frac{4N\log T}{\min_{i\in[N]}\{T_{i,k}(t)\}}}\right) \leq \frac{2}{T^2}$$

which ends the proof of Lemma A.1 □

We provide lemmas for the convergence bound of randomised gossip algorithms. Similar proof could be found Lei et al. [2020], Achddou et al. [2024].

**Lemma A.3** (Random Graph). *Let us assume that the communication graph follows Assumption 3.1. For $t = 1 \ldots T$, $W_t$ is doubly stochastic matrix and symmetric and i.i.d. Then $\forall v \in V$, $\forall s, t \in [T]$ such that $t > s$,*

$$\mathbb{P}\left(\left\|W_t \cdots W_{s+1}e_v - \frac{1}{N}\mathbf{1}\right\|_2 \geq \delta\right) \leq \frac{\lambda_2(\mathbb{E}[W^2])^{t-s}}{\delta^2}.$$

*When $t - s \geq \left\lceil\frac{3\log(T)}{\log\lambda_2(\mathbb{E}[W^2])^{-1}}\right\rceil = \tau'$, we have*

$$\mathbb{P}\left(\left\|W_t \cdots W_{s+1}e_v - \frac{1}{N}\mathbf{1}\right\|_2 \geq \delta\right) \leq \delta. \tag{10}$$

*Furthermore, when $t - s \geq \tau^* = \left\lceil\frac{3N\log(T)}{p\lambda_{N-1}(\mathrm{Lap}(\mathcal{G}))}\right\rceil \geq \tau'$, Equation (10) still holds.*

*Proof.* Using Markov's inequality we have

$$\mathbb{P}\left(\left\|W_t\cdots W_{s+1}e_v - \frac{1}{N}\mathbf{1}\right\|_2 \geq \delta\right) \leq \frac{\mathbb{E}\left(\left\|W_t\cdots W_{s+1}e_v - \frac{1}{N}\mathbf{1}\right\|_2^2\right)}{\delta^2}.$$

Let $\tilde{W}_k = W_k - \frac{1}{N}\mathbf{1}\mathbf{1}^\top$ and assume

$$\mathbb{E}\left[\left\|W_{k-1}\cdots W_{s+1}e_v - \frac{1}{N}\mathbf{1}\right\|_2^2\right] \leq e_v^T e_v \left\|\mathbb{E}[W_1 W_1^\top] - \frac{1}{N}\mathbf{1}\mathbf{1}^\top\right\|_{\text{op}}^{k-s-1} \qquad \text{for some } k-1 > s.$$

Let $\mathcal{F}_{k-1}$ be the $\sigma$-algebra generated by all random events up to time $k-1$. We have that

$$\mathbb{E}\left[\left\|W_k^\top\cdots W_{s+1}^\top e_v - \frac{1}{N}\mathbf{1}\right\|_2^2\right] = \mathbb{E}\left[e_v^T \tilde{W}_{s+1}^\top \cdots \tilde{W}_{k-1}^\top \tilde{W}_k^\top \tilde{W}_k \tilde{W}_{k-1}\cdots \tilde{W}_{s+1}e_v\right]$$

$$= \mathbb{E}\left[e_v^T \tilde{W}_{s+1}^\top \cdots \tilde{W}_{k-1}^\top \mathbb{E}[\tilde{W}_k^\top \tilde{W}_k \mid \mathcal{F}_{k-1}]\tilde{W}_{k-1}\cdots \tilde{W}_{s+1}e_v\right]$$

$$= \mathbb{E}\left[e_v^T \tilde{W}_{s+1}^\top \cdots \tilde{W}_{k-1}^\top \mathbb{E}[\tilde{W}_1^\top \tilde{W}_1]\tilde{W}_{k-1}\cdots \tilde{W}_{s+1}e_v\right]$$

$$\hspace{6cm}\text{(by independence of } W_k)$$

$$\leq \left\|\mathbb{E}[W_1 W_1^\top] - \frac{1}{N}\mathbf{1}\mathbf{1}^\top\right\|_{\text{op}} e_v^T e_v \left\|\mathbb{E}[W_1 W_1^\top] - \frac{1}{N}\mathbf{1}\mathbf{1}^\top\right\|_{\text{op}}^{k-s-1}$$

$$\leq \lambda_2(\mathbb{E}[W^2])^{t-s} e_v^T e_v \left\|\mathbb{E}[W_1 W_1^\top] - \frac{1}{N}\mathbf{1}\mathbf{1}^\top\right\|_{\text{op}}^{k-s-1}$$

which by induction, suffices to prove the lemma. Moreover, the proof of $\tau^* \geq \tau'$ follows from the fact that

$$\frac{1}{log\lambda_2^{-1}} \leq \frac{1}{1-\lambda_2}, \lambda_2(\mathbb{E}[W^2]) \leq 1 - \frac{p\lambda_{N-1}(\text{Lap}(\mathcal{G}))}{N} \text{ and } \frac{1}{\log(1-p)^{-1}} \leq \frac{1}{p},$$

where the second inequality is taken from Theorem 6.1 in Achddou et al. [2024]. $\qquad\square$

The following lemmas guarantee local consistency between agents.

**Lemma A.4.** *Let us assume that the communication graph follows Assumption 3.1. Then with probability $1 - N^2 T\delta$, Algorithm 1 guarantees that for fixed arm $k \in [K]$, and for every $i, j \in [N]$ and for every $t \in [T]$,*

$$|T_{i,k}(t) - T_{j,k}(t)| \leq KN L_p(\delta)$$

*where $L_p(\delta) = \left\lceil\frac{\log(\delta)}{\log(1-p)}\right\rceil$ denotes the maximum number of rounds each edge within base graph $\mathcal{G}$ is connected in the communication graph $\mathcal{G}_t$ with probability $1 - \delta$.*

*Proof.* Fix an agent $i \in [N]$ and a time step $t \in [T]$. According to Assumption 3.1 and Algorithm 1, $p$ be the probability that agent $i$ communicates with a fixed neighbour $j \in \mathcal{N}_i(t)$ in any given step, independently of the past. For a non-negative integer $L$, we have

$$\mathbb{P}\left(i \text{ does not contact j during the next } L+1 \text{ steps}\right) = (1-p)^{L+1}.$$

The communication gap length at time $t$ is the number of steps starting from time $t$ until agent $i$ next successfully communicates with agent $j$. Choose a confidence parameter $\delta \in (0,1)$, we have

$$\mathbb{P}\left(\text{time until first contact between agents } i \text{ and } j \text{ exceeds } L_p(\delta)\right) \leq \delta.$$

Applying a union bound to gives

$$\mathbb{P}\left(\exists i \in [N], j \in \mathcal{N}_i, t \in [T] : \text{time until first contact between } i \text{ and } j \text{ after time } t \text{ exceeds } L_p(\delta)\right)$$
$$\leq N^2 T\delta.$$

Note that we have $\mathcal{N}_i(t) \subseteq \mathcal{N}_i$, where $\mathcal{N}_i$ is a fixed superset of possible neighbors. Hence with probability at least $1 - N^2 T \delta$, for every agent $i \in [N]$, every time step $t \in [T]$, and every neighbor $j \in \mathcal{N}_i$, the time until the next communication between $i$ and $j$ is at most $L_p(\delta)$ time steps.

For any two agents $i$ and $j$, let $d(i,j) \leq N$ denote the shortest path length between $i$ and $j$ in the base graph $\mathcal{G}$. Because with high probability, information can gossip across each edge within at most $L_p(\delta)$ steps, it follows that information originating at $i$ at time step $t$ reaches $j$ at most by time step

$$t + L_p(\delta) \cdot d(i,j) \leq t + L_p(\delta) \cdot N. \tag{11}$$

In order to be cautious for notations, we use $\mathcal{S}_i(t)$ to denote the active set in round $t$ for Algorithm 1. According to Algorithm 1, during every communication, each agent $i$ replaces its active set by the intersection with its neighbors:

$$\mathcal{S}_i(t+1) = \bigcap_{j \in \mathcal{N}_i(t) \cup \{i\}} \mathcal{S}_j(t).$$

because for all $\mathcal{S}_i(0) = [K]$, at most $K$ distinct arms can ever be removed. Each arm can start a new wave of disagreement. Each wave of disagreement at most last $L_p(\delta) \cdot N$. In the worst case, the waves do not overlap. Consequently the longest possible sequence of rounds in which $\mathcal{S}_i(t) \neq \mathcal{S}_j(t)$ is $K \cdot N \cdot L_p(\delta)$. During the disagreement period, for a fixed arm $k$ we could upper bound of maximum pull of $k$ is $KNL_p(\delta)$ and lower bound of maximum pull of $k$ is 0. Hence we have

$$|T_{i,k}(t) - T_{j,k}(t)| \leq KNL_p(\delta).$$

$\square$

# B   Omitted details in Section 5

In this section, we show the omitted details in Section 5.

*Proof of Lemma 5.1.* When $T_{i,k}(t) \leq KL^*$, the bound trivially holds.

Now we consider the case when $T_{i,k}(t) > KL^*$, We first define the following event:

$$E := \left\{ \bigcap_{\substack{k \in [K] \\ t \in [T]}} \left\{ \frac{\left| \sum_{j \in [N]} (\widehat{\mu}_{j,k}(t) - \mu_{j,k}) \right|}{N} \leq \sqrt{\frac{4 \log(T)}{N \min_{j \in [N]} \{T_{j,k}(t)\}}} \right\} \right\}$$

$$\cap \left\{ \bigcap_{\substack{j \in [N] \\ \tau^* \leq t - s \leq T}} \left\| W_t \cdots W_{s+1} e_j - \frac{1}{N} \mathbf{1} \right\|_2 \leq \frac{1}{T^2} \right\} \cap \left\{ \bigcap_{\substack{i,j \in [N] \\ k \in [K] \\ t \in [T]}} |T_{i,k}(t) - T_{j,k}(t)| \leq KL^* \right\}. \tag{12}$$

In Equation (12), the first event bounds the global estimation error for each arm, the second ensures near-uniform information mixing across agents after sufficient communication rounds, and the third guarantees that the number of pulls for any arm remains approximately balanced across agents at each time step. By applying Lemma A.2, A.3, A.4 and union bound, we obtain that $\mathbb{P}[E^c] \leq \frac{K}{T} + \frac{N}{T} + \frac{1}{T} \leq \frac{3NK}{T}$ when we set $\delta = \frac{1}{N^2 T^2}$ and $L^* = NL_p\left(\frac{1}{N^2 T^2}\right) = N \left\lceil -\frac{2 \log(NT)}{\log(1-p)} \right\rceil$.

We define $\widetilde{\mu}_k(t) = \frac{1}{N} \sum_{j \in [N]} z_{j,k}(t)$ to be an intermediate variable that has access to each agent's average mean on arm $k$ at time $t$.

For any agent $i$, we have

$$\begin{aligned} |z_{i,k}(t) - \mu_k| &= |z_{i,k}(t) - \widetilde{\mu}_k(t) + \widetilde{\mu}_k(t) - \mu_k| \\ &\leq |\widetilde{\mu}_k(t) - z_{i,k}(t)| + |\widetilde{\mu}_k(t) - \mu_k| \end{aligned} \qquad \text{(Triangle inequality)}$$

$$= \underbrace{|\widetilde{\mu}_k(t) - z_{i,k}(t)|}_{\text{Consensus Error}} + \underbrace{\frac{\left|\sum_{j\in[N]}(\widehat{\mu}_{j,k}(t) - \mu_{j,k})\right|}{N}}_{\text{Estimation Error}}, \tag{13}$$

The last equality is due to the definition of the global mean reward and $\widetilde{\mu}_k(t)$. Let us first focus on Consensus Error. For any $i \in [N]$, According to the update in Equation (4) we have

$$
\begin{aligned}
z_{i,k}(t) &= \sum_{j\in[N]} [W_{t-1}]_{i,j} z_{i,k}(t-1) + \widehat{\mu}_{i,k}(t-1) - \widehat{\mu}_{i,k}(t-2) \\
&= \sum_{j\in[N]} [W_{t-1}\cdots W_{t-s}]_{i,j} z_{i,k}(t-s) + \sum_{\tau=t-s}^{t-2} \sum_{j\in[N]} [W_{t-2}\cdots W_{\tau+1}]_{i,j} (\widehat{\mu}_{j,k}(\tau) - \widehat{\mu}_{j,k}(\tau-1)) \\
&\quad + \widehat{\mu}_{i,k}(t-1) - \widehat{\mu}_{i,k}(t-2) \tag{14} \\
&= \sum_{j\in[N]} [W_{t-1}\cdots W_1]_{i,j} z_{i,k}(1) + \sum_{\tau=1}^{t-2} \sum_{j\in[N]} [W_{t-2}\cdots W_{\tau+1}]_{i,j} (\widehat{\mu}_{j,k}(\tau) - \widehat{\mu}_{j,k}(\tau-1)) \\
&\quad + \widehat{\mu}_{i,k}(t-1) - \widehat{\mu}_{i,k}(t-2). \qquad\qquad (\text{setting } s = t-1)
\end{aligned}
$$

We also have

$$
\begin{aligned}
\widetilde{\mu}_k(t) &= \frac{1}{N} \sum_{j\in[N]} z_{j,k}(t) \\
&= \widetilde{\mu}_k(t-s) + \frac{1}{N} \sum_{\tau=t-s}^{t-1} \sum_{j\in[N]} (\widehat{\mu}_{j,k}(\tau) - \widehat{\mu}_{j,k}(\tau-1)) \\
&= \widetilde{\mu}_k(1) + \frac{1}{N} \sum_{\tau=1}^{t-1} \sum_{j\in[N]} (\widehat{\mu}_{j,k}(\tau) - \widehat{\mu}_{j,k}(\tau-1)) \qquad (\text{setting } s = t-1) \\
&= \frac{1}{N} \sum_{j\in[N]} z_{j,k}(1) + \frac{1}{N} \sum_{\tau=1}^{t-1} \sum_{j\in[N]} (\widehat{\mu}_{j,k}(\tau) - \widehat{\mu}_{j,k}(\tau-1)) \\
&= \frac{1}{N} \sum_{j\in[N]} z_{j,k}(1) + \frac{1}{N} \sum_{\tau=1}^{t-2} \sum_{j\in[N]} (\widehat{\mu}_{j,k}(\tau) - \widehat{\mu}_{j,k}(\tau-1)) \\
&\quad + \frac{1}{N} \sum_{j\in[N]} (\widehat{\mu}_{j,k}(t-1) - \widehat{\mu}_{j,k}(t-2)) \tag{15}
\end{aligned}
$$

Hence, we obtain

$$
\begin{aligned}
\widetilde{\mu}_k(t) - z_{i,k}(t) &= \sum_{\tau=1}^{t-2} \left( \sum_{j\in[N]} \left(\frac{1}{N} - [W_{t-2}\cdots W_{\tau+1}]_{i,j}\right) (\widehat{\mu}_{j,k}(\tau) - \widehat{\mu}_{j,k}(\tau-1)) \right) \\
&\quad + \frac{1}{N} \sum_{j\in[N]} (\widehat{\mu}_{j,k}(t-1) - \widehat{\mu}_{j,k}(t-2)) - (\widehat{\mu}_{i,k}(t-1) - \widehat{\mu}_{i,k}(t-2)) \\
&\quad + \frac{1}{N} \sum_{j\in[N]} z_{j,k}(1) - \sum_{j\in[N]} [W_{t-1}\cdots W_1]_{i,j} z_{j,k}(1). \\
&= \sum_{\tau=1}^{t-2} \left( \sum_{j\in[N]} \left(\frac{1}{N} - [W_{t-2}\cdots W_{\tau+1}]_{i,j}\right) (\widehat{\mu}_{j,k}(\tau) - \widehat{\mu}_{j,k}(\tau-1)) \right) \\
&\quad + \frac{1}{N} \sum_{j\in[N]} (\widehat{\mu}_{j,k}(t-1) - \widehat{\mu}_{j,k}(t-2)) - (\widehat{\mu}_{i,k}(t-1) - \widehat{\mu}_{i,k}(t-2)),
\end{aligned}
$$

.Taking absolute values on both sides, we have

$$|\widetilde{\mu}_k(t) - z_{i,k}(t)| \leq \left| \sum_{\tau=1}^{t-2} \left( \sum_{j\in[N]} \left( \frac{1}{N}[W_{t-2}\cdots W_{\tau+1}]_{i,j} \right)(\widehat{\mu}_{j,k}(\tau) - \widehat{\mu}_{j,k}(\tau-1)) \right) \right|$$

$$+ \left| \frac{1}{N}\sum_{j\in[N]}(\widehat{\mu}_{j,k}(t-1) - \widehat{\mu}_{j,k}(t-2)) - (\widehat{\mu}_{i,k}(t-1) - \widehat{\mu}_{i,k}(t-2)) \right|$$

$$\leq \underbrace{\left| \sum_{\tau=1}^{t-\tau^*-2} \left( \sum_{j\in[N]} \left( \frac{1}{N} - [W_{t-2}\cdots W_{\tau+1}]_{i,j} \right)(\widehat{\mu}_{j,k}(\tau) - \widehat{\mu}_{j,k}(\tau-1)) \right) \right|}_{\heartsuit}$$

$$+ \underbrace{\left| \sum_{\tau=t-\tau^*-1}^{t-2} \left( \sum_{j\in[N]} \left( \frac{1}{N} - [W_{t-2}\cdots W_{\tau+1}]_{i,j} \right)(\widehat{\mu}_{j,k}(\tau) - \widehat{\mu}_{j,k}(\tau-1)) \right) \right|}_{\spadesuit}$$

$$+ \underbrace{\left| \frac{1}{N}\sum_{j\in[N]}(\widehat{\mu}_{j,k}(t-1) - \widehat{\mu}_{j,k}(t-2)) - (\widehat{\mu}_{i,k}(t-1) - \widehat{\mu}_{i,k}(t-2)) \right|}_{\clubsuit}. \quad (16)$$

Now we analyze three terms on the right-hand side of Equation (16).

**Bounding term $\heartsuit$.** Conditioning on event $E$, we obtain

$$\heartsuit = \left| \sum_{\tau=1}^{t-\tau^*-2} \left( \sum_{j\in[N]} \left( \frac{1}{N} - [W_{t-2}\cdots W_{\tau+1}]_{i,j} \right)(\widehat{\mu}_{j,k}(\tau) - \widehat{\mu}_{j,k}(\tau-1)) \right) \right|$$

$$\leq \sum_{\tau=1}^{t-\tau^*-2} \sum_{j\in[N]} \left| \frac{1}{N} - [W_{t-2}\cdots W_{\tau+1}]_{i,j} \right| \qquad \text{(rewards are bounded in the interval } [0,1])$$

$$= \sum_{\tau=1}^{t-\tau^*-2} \left\| W_{t-2}\cdots W_{\tau+1}e_i - \frac{\mathbf{1}}{N} \right\|_1$$

$$\leq \sqrt{N} \cdot \sum_{\tau=1}^{t-\tau^*-2} \left\| W_{t-2}\cdots W_{\tau+1}e_i - \frac{\mathbf{1}}{N} \right\|_2$$

$$\leq \frac{\sqrt{N}(t - \tau^*)}{T^2}$$

$$\leq \frac{\sqrt{N}}{T}$$

$$\leq \frac{\sqrt{N}}{T_{i,k}(t)}, \quad (17)$$

where the third inequality comes from the condition that $t - \tau - 1 \in [\tau^*, t-3]$ and the event $E$.

**Bounding term $\spadesuit$.** We have

$$\spadesuit = \left| \sum_{\tau=t-\tau^*-1}^{t-2} \left( \sum_{j\in[N]} \left( \frac{1}{N} - [W_{t-1}\cdots W_{\tau+1}]_{i,j} \right)(\widehat{\mu}_{j,k}(\tau) - \widehat{\mu}_{j,k}(\tau-1)) \right) \right|$$

$$\leq \sum_{\tau=t-\tau^*-1}^{t-2} \left( \sum_{j\in[N]} \left| \left( \frac{1}{N} - [W_{t-1}\cdots W_{\tau+1}]_{i,j} \right) \right| \right) \cdot$$

$$\left| \frac{\sum_{s=1}^{\tau} \mathbb{I}\{A_j(s) = k\} X_{j,k}(s)}{T_{j,k}(\tau)} - \frac{\sum_{s=1}^{\tau-1} \mathbb{I}\{A_j(s) = k\} X_{j,k}(s)}{T_{j,k}(\tau - 1)} \right| \right) \qquad (\text{ definition of } \widehat{\mu}_{j,k}(t))$$

$$\leq \sum_{\tau=t-\tau^*-1}^{t-2} \left( \sum_{j \in [N]} \left| \left( \frac{1}{N} - [W_{t-1} \cdots W_{\tau+1}]_{i,j} \right) \right| \cdot \right.$$
$$\left. \left| \frac{\sum_{s=1}^{\tau-1} \mathbb{I}\{A_j(s) = k\} X_{j,k}(s) + X_{j,k}(\tau)}{T_{j,k}(\tau)} - \frac{\sum_{s=1}^{\tau-1} \mathbb{I}\{A_j(s) = k\} X_{j,k}(s)}{T_{j,k}(\tau) - 1} \right| \right)$$

$$\leq \sum_{\tau=t-\tau^*-1}^{t-2} \left( \sum_{j \in [N]} \left| \left( \frac{1}{N} - [W_{t-1} \cdots W_{\tau+1}]_{i,j} \right) \right| \cdot \right.$$
$$\left. \left| \frac{-\sum_{s=1}^{\tau-1} \mathbb{I}\{A_j(s) = k\} X_{j,k}(s) + (T_{j,k}(\tau) - 1) X_{j,k}(\tau)}{T_{j,k}(\tau) \, (T_{j,k}(\tau) - 1)} \right| \right)$$

$$\leq \sum_{\tau=t-\tau^*-1}^{t-2} \sum_{j \in [N]} \left| \left( \frac{1}{N} - [W_{t-1} \cdots W_{\tau+1}]_{i,j} \right) \right| \frac{1}{T_{j,k}(\tau)}$$
$$(\text{rewards are bounded in the interval } [0,1])$$

$$\leq \sum_{\tau=t-\tau^*-1}^{t-2} \sum_{j \in [N]} \left| \left( \frac{1}{N} - [W_{t-1} \cdots W_{\tau+1}]_{i,j} \right) \right| \frac{1}{\max \{T_{i,k}(\tau) - KL^*, 1\}} \qquad (\text{event } E)$$

$$\leq \frac{4\tau^*}{\max \{T_{i,k}(t) - KL^*, 1\}},$$

where the second inequality follows from the fact that there are only two possible cases for $T_{j,k}(\tau)$ and $T_{j,k}(\tau - 1)$. When $T_{j,k}(\tau) = T_{j,k}(\tau - 1)$ the absolute difference of means is trivially 0, so the only non-trivial case to consider is when $T_{j,k}(\tau) = T_{j,k}(\tau - 1) + 1$. The last inequality is due to that the upper bound of the total-variation distance from any distribution to the uniform distribution is 1.

**Bounding term ♣.** Conditioning on $E$, we obtain

$$\clubsuit = \left| \frac{1}{N} \sum_{j \in [N]} (\widehat{\mu}_{j,k}(t-1) - \widehat{\mu}_{j,k}(t-2)) - (\widehat{\mu}_{i,k}(t-1) - \widehat{\mu}_{i,k}(t-2)) \right|$$

$$\leq \sum_{j \in [N]} \frac{2}{N T_{j,k}(t-1)} + \frac{2}{T_{i,k}(t-1)}$$

$$\leq \frac{4}{\max \{T_{i,k}(t-1) - KL^*, 1\}},$$

Next, let us analyse Estimation Error. Conditioned on $E$, for all $k \in [K]$ and $t \in [T]$ we obtain

$$\frac{\left| \sum_{j \in [N]} (\widehat{\mu}_{j,k}(t) - \mu_{j,k}) \right|}{N} \leq \sqrt{\frac{4 \log(T)}{N \min_{j \in [N]} \{T_{j,k}(t)\}}}$$

$$\leq \sqrt{\frac{4 \log(T)}{N \max \{T_{i,k}(t) - KL^*, 1\}}}, \qquad (18)$$

where the last inequality is due to the event $E$.

Combining all the results collected so far, we can finally derive the concentration bound conditioned on the event $E$. For any $i \in [N]$ and $k \in [K]$, we obtain

$$|z_{i,k}(t) - \mu_k| \leq \sqrt{\frac{4 \log(T)}{N \max \{T_{i,k}(t) - KL^*, 1\}}} + \frac{4 \left( \sqrt{N} + \tau^* \right)}{\max \{T_{i,k}(t) - KL^*, 1\}}. \qquad (19)$$

$\square$

*Proof of Lemma 5.2.* First, for all agents we consider the cases under the event $E$. According to Algorithm 1, if arm $k$ is eliminated, there are only two possible cases: 1) there exists some $k'$ such that $z_{i,k}(t) + c_{i,k}(t) \leq z_{i,k'}(t) - c_{i,k'}(t)$; 2) When Algorithm 1 updates the active set $\mathcal{S}_i(t+1) \leftarrow \bigcap_{j \in \mathcal{N}_i(t) \cup \{i\}} \mathcal{S}_j(t)$, $k \notin \mathcal{S}_j$ for any $j \in \mathcal{N}_i(t)$.

For Case 1, Since we have $z_{i,k}(t) + c_{i,k}(t) \leq \mu_k + 2c_{i,k}(t)$ as well as $z_{i,k'}(t) - c_{i,k'}(t) \geq \mu_{k'} - 2c_{i,k'}(t)$, then when

$$2(c_{i,k}(t) + c_{i,k'}(t)) \leq \mu_{k'} - \mu_k \leq \Delta_k$$

arm $k$ will be essentially eliminated. Due to the pulling rule of Algorithm 1, we have $|T_{i,k}(t) - T_{i,k'}(t)| \leq 1$. Thus when

$$\Delta_k \geq 2 \left( \sqrt{\frac{4\log(T)}{N \max\{T_{i,k}(t) - KL^*, 1\}}} + \frac{4\left(\sqrt{N} + \tau^*\right)}{\max\{T_{i,k}(t) - KL^*, 1\}} \right).$$

Hence, here we know that when

$$T_{i,k}(t) \geq \frac{64\log(T)}{N\Delta_k^2} + \frac{16\left(\sqrt{N} + \tau^*\right)}{\Delta_k} + KL^*,$$

arm $k$ will be essentially eliminated.

For Case 2, the optimal arm $k^*$ it cannot be eliminated, because for agents $i$ we consider the cases under the event $E$.

For all agents cases under event $E^c$, we have

$$T \cdot \mathbb{P}(E^c)\Delta_{\max} \leq 3KN\Delta_{\max}.$$

Therefore, combining all inequalities above, we have

$$\text{Reg}_{i,T}(\pi) \leq \sum_{k:\Delta_k > 0} \left( \frac{64\log(T)}{N\Delta_k} + 16\left(\sqrt{N} + \tau^*\right) \right) + KL^* + 3KN\Delta_{\max}$$

which ends the proof. $\square$

*Proof of Theorem 5.3.* By adding up Lemma 5.2 for all agents $i \in \mathcal{N}$, Theorem 5.3 can be proved through the facts that

$$\frac{1}{\log\lambda_2^{-1}} \leq \frac{1}{1 - \lambda_2}, \lambda_2(\mathbb{E}[W^2]) \leq 1 - \frac{p\lambda_{N-1}(\text{Lap}(\mathcal{G}))}{N} \text{ and } \frac{1}{\log(1-p)^{-1}} \leq \frac{1}{p},$$

where the second inequality is taken from Theorem 6.1 in Achddou et al. [2024]. $\square$

*Proof of Corollary 5.4.* If $\mathcal{G}$ is complete, then $d(i,j) = 1$ for all $i, j \in \mathcal{N}$, and the gossip time simplifies to $t + L$. Recall Equation (11), with the same steps we can prove that

$$|T_{i,k}(t) - T_{j.k}(t)| \leq KL_p(\delta) = K \left\lceil \frac{\log(\delta)}{\log(1-p)} \right\rceil$$

for any arm $k \in [K]$ and agents $i, j \in [N]$ for any $t \in [T]$.

Hence by the fact that $\lambda_{N-1}(\text{Lap}(\mathcal{G})) = N$ and let $L^*$ as defined in Corollary 5.4 we can prove the case for complete graph.

For the rest two cases, we can just prove which by substituting the exact values of $\lambda_{N-1}(\text{Lap}(\mathcal{G}))$ in Theorem 5.3. $\square$

*Proof of Theorem 5.9.* First, note that

$$\text{Reg}_T^\nu(\pi) = \sum_{i \in [N]} \text{Reg}_{i,T}^\nu(\pi) = \sum_{i \in [N]} \sum_{k \in [K]} \mathbb{E}[T_{i,k}(T)]\Delta_k = \sum_{k \in [K]} \Delta_k \sum_{i \in [N]} \mathbb{E}[T_{i,k}(T)].$$

In order to show Theorem 5.9, we only need to prove that

$$\lim_{T\to\infty} \frac{\sum_{i\in[N]} \mathbb{E}[T_{i,k}(T)]}{\log T} \geq \frac{2(1-s)}{(1+\varepsilon)^2 \Delta_k^2} \tag{20}$$

for $p$ and $\varepsilon$ and some sub-optimal arm $k \neq k^*$.

Suppose the distribution over instance $\nu$ is given by $\mathcal{P} = (\mathbb{P}_{i,a})_{i\in[N],a\in[K]}$. Consider another instance $\nu'$ with $\mathcal{P}' = (\mathbb{P}'_{i,a})_{i\in[N],a\in[K]}$ such that

$$\mathbb{P}'_{i,a} = \begin{cases} \mathcal{N}(\mu_{i,a}, 1), \ a \neq k \\ \mathcal{N}(\mu_{i,a} + (1+\varepsilon)\Delta_a, 1), \ a = k \end{cases}$$

for $i \in [N]$ where $\mathbb{P}_{i,a} = \mathcal{N}(\mu_{i,a}, 1)$ and $\varepsilon \in (0,1]$.

According to the consistency of policy $\pi$, it holds that

$$\text{Reg}_T^\nu(\pi) + \text{Reg}_T^{\nu'}(\pi) \leq 2CT^s$$

for some constant $C$ and $p \in (0,1)$.

Simultaneously, let event $A_i = \left\{ T_{i,k}(T) \geq \frac{T}{2} \right\}$, then

$$\begin{aligned}
\text{Reg}_{i,T}^\nu(\pi) + \text{Reg}_{i,T}^{\nu'}(\pi) &\geq \frac{T}{2} \cdot \Delta_k \cdot \mathbb{P}_{\nu,\pi}[A_i] + \frac{T}{2} \cdot \varepsilon \cdot \Delta_k \cdot \mathbb{P}_{\nu',\pi}[A_i^c] \\
&\geq \frac{T}{2} \varepsilon \Delta_k \left( \mathbb{P}_{\nu,\pi}[A_i] + \mathbb{P}_{\nu',\pi}[A_i^c] \right) \\
&\geq \frac{T}{4} \varepsilon \Delta_k \exp\left( -\text{KL}(\nu, \nu') \right) \\
&= \frac{T}{4} \varepsilon \Delta_k \exp\left( \sum_{j\in[N]} \mathbb{E}[T_{j,k}(T)] \cdot \text{KL}(\mathbb{P}_{j,a}, \mathbb{P}'_{j,a}) \right) \\
&= \frac{T}{4} \varepsilon \Delta_k \exp\left( -\sum_{j\in[N]} \mathbb{E}[T_{j,k}(T)] \cdot \frac{(1+\varepsilon)^2 \Delta_k^2}{2} \right).
\end{aligned}$$

Hence, summing the above inequality for all agents $i \in [N]$, we obtain

$$\text{Reg}_T^\nu(\pi) + \text{Reg}_T^{\nu'}(\pi) \geq \frac{NT}{4} \varepsilon \Delta_k \exp\left( -\sum_{j\in[N]} \mathbb{E}[T_{j,k}(T)] \cdot \frac{(1+\varepsilon)^2 \Delta_k^2}{2} \right). \tag{21}$$

Rearranging the terms in Equation (21), it holds that

$$\begin{aligned}
\sum_{j\in[N]} \mathbb{E}[T_{j,k}(T)] \cdot \frac{(1+\varepsilon)^2 \Delta_k^2}{2} &\geq \log\left( \frac{NT\varepsilon\Delta_k/4}{\text{Reg}_T^\nu(\pi) + \text{Reg}_T^{\nu'}(\pi)} \right) \\
&\geq \log\left( \frac{NT\varepsilon\Delta_k}{8CT^s} \right) = (1-s)\log(T) + \log\left( \frac{N\varepsilon\Delta_k}{8C} \right).
\end{aligned}$$

Therefore, we can show

$$\lim_{T\to\infty} \frac{\sum_{i\in[N]} \mathbb{E}[T_{j,k}(T)]}{\log T} \geq \frac{2(1-s)}{(1+\varepsilon)^2 \Delta_k^2}$$

which is the goal in Equation (20). $\qquad\square$

## C  Estimation of unknown link probability

Since $\mathcal{G}$ is connected, each agent has at least one neighbor. This allows every agent to estimate the edge activation probability $p$ by observing its connectivity status over multiple rounds. We design the following procedure for each agent $i \in [N]$ to compute an estimate $\hat{p}$ of $p$.

---

**Algorithm 2** Burn-in Phase for Estimating $p$ by Agent $i \in [N]$

---

1: **Input:** Confidence level $\delta \in (0,1)$, any fixed neighbor $n_i \in [N]_i$ of agent $i$ in the base graph $\mathcal{G}$
2: **Initialization:** Set $\tilde{\tau}_i \leftarrow 0$, $t \leftarrow 0$, $\hat{p}_i(0) \leftarrow 0$ and $\mathrm{CI}_i(0) \leftarrow \infty$
3: **while** $\hat{p}_i(t) - 3\mathrm{CI}_i(t) \leq 0$ **do**
4:     Increment time $t \leftarrow t + 1$ and observe $\mathcal{N}_i(t)$
5:     **if** $n_i \in \mathcal{N}_i(t)$ **then**
6:         $\tilde{\tau}_i \leftarrow \tilde{\tau}_i + 1$
7:     **end if**
8:     Select an arm uniformly at random
9:     Update $\hat{p}_i(t) = \frac{\tilde{\tau}_i}{t}$ and $\mathrm{CI}_i(t) = \sqrt{\frac{\log(2/\delta)}{2t}}$
10: **end while**
11: **Output:** $\hat{p}_i = \hat{p}_i(t) - \mathrm{CI}_i(t)$

---

**Lemma C.1.** *For the agent $i$ and each $t$, it holds that*

$$\mathbb{P}\left[|\hat{p}_i(t) - p| \leq \mathrm{CI}_i(t)\right] \geq 1 - \delta.$$

*Proof of Lemma C.1.* Lemma C.1 comes directly from Hoeffding's inequality.     □

**Theorem C.2.** *Let $\delta = \frac{2}{T^2}$. Then, with probability at least $1 - \frac{2N}{T}$, the estimate $\hat{p}_i$ returned by Algorithm 2 satisfies $\hat{p}_i \in \left(\frac{p}{2}, p\right]$ for every agent $i \in [N]$. Furthermore, the cumulative regret incurred during the burn-in phase across all agents is bounded by $\mathcal{O}\left(\frac{N \log T}{p^2}\right)$.*

*Proof of Theorem C.2.* Suppose the end round for Algorithm 2 is $t^*$. As $\hat{p}_i = \hat{p}_i(t) - \mathrm{CI}_i(t) \leq p$ according to Lemma C.1, we only need to show $\hat{p}_i > \frac{p}{2}$. This can be verified by

$$\hat{p}_i = \hat{p}_i(t) - \mathrm{CI}_i(t) > \frac{\hat{p}_i(t) + \mathrm{CI}_i(t)}{2} \geq \frac{p}{2}.$$

By combining each $t \in [T]$ and $i \in [N]$ and union bound, we can obtain the first part of Theorem C.2. Moreover, for $t^* > \frac{16 \log(T)}{p^2}$, we have

$$\hat{p}_i(t) - 3\mathrm{CI}_i(t) \geq p - 4\mathrm{CI}_i(t) = p - 4\sqrt{\frac{\log T}{t^*}} > 0,$$

hence the stopping condition in Algorithm 2 holds. By the fact that $\Delta_{\max} \leq 1$ for all arms, we can obtain the second part of Theorem C.2 through adding the regret for all agents $i \in [N]$.     □

# D   Additional experiments

To further validate the dependence of the regret on the link probability $p$ and algebraic connectivity $\lambda_{N-1}(\mathrm{Lap}(\mathcal{G}))$, we run the following additional experiments.

**Experiment on Link Probability $p$.**   We conduct additional experiments on complete base graphs with $p \in [0.04, 0.18]$, and report the log-log relationship between $p$ and the resulting regret. The results are shown in Figure 2. By performing a linear regression between $\log(p)$ and $\log(\mathrm{Regret})$, we obtain a slope $\hat{\alpha} = -0.93$ with $R^2 = 1.0$, indicating a nearly perfect linear fit (of order $\frac{1}{p^{0.93}}$). Surprisingly, we also report the $R^2$ corresponding to the curve of fit $\frac{1}{p}$, and find out that $R^2$ is $0.995$, which represents that the curve fit is quite statistically significant and thus a perfect linear fit. This strongly supports the inverse proportionality between regret and $p$, i.e., regret scales approximately as $O(1/p)$, which is consistent with our theoretical upper bound. This result empirically also validates that $p$ leads to significantly higher regret due to slower information diffusion across agents.

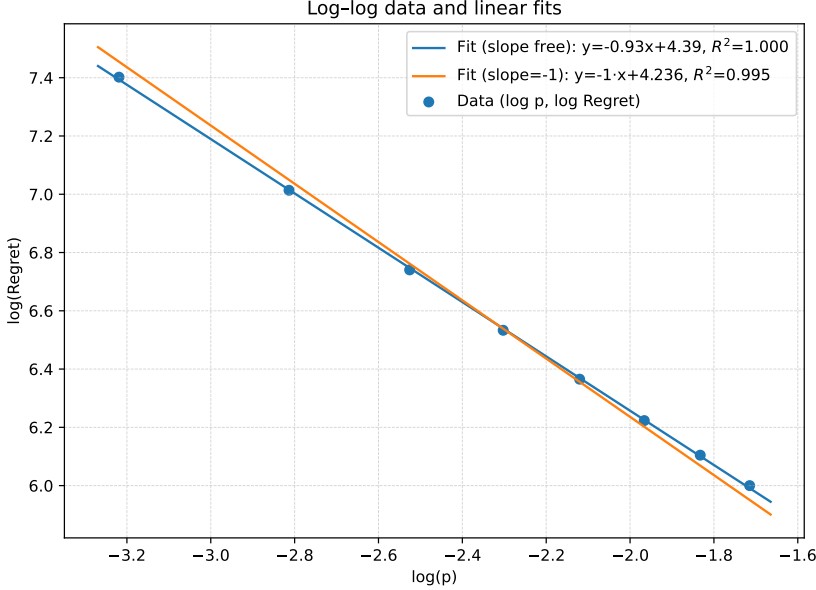

Figure 2: Log-log data for GSE: $\log(p)$ and $\log(\text{Regret})$ with classical ER graph, $p = 0.9$)

**Experiment on Algebraic Connectivity** $\lambda_{N-1}(\text{Lap}(\mathcal{G}))$. We conduct experiments using $d$-regular graphs with varying degrees $d$. Specifically, we construct the base graphs as circulant graphs. This structure provides a controllable family of regular graphs with increasing algebraic connectivity as $d$ grows. We run our algorithm GSE under this setting with the link probability $p = 0.9$. Table 1 shows a clear inverse relationship: as $\lambda_{N-1}(\text{Lap}(\mathcal{G}))$ increases with higher $d$, the regret decreases significantly. This supports our theoretical findings that larger algebra connectivity leads to more efficient information propagation and thus lower regret.

Table 1: Average regret of GSE with $d$-regular graph

| $d$-regular graph | 2 | 4 | 6 | 8 | 10 | 12 | 14 |
|---|---|---|---|---|---|---|---|
| $\lambda_{N-1}(\mathcal{G})$ | 0.1716 | 1.39 | 1.97 | 3.97 | 6.73 | 10.15 | 14.00 |
| Regret | 751.28 | 255.52 | 165.66 | 133.92 | 121.93 | 115.12 | 112.47 |

# E  Notations

The following notation chart provide notation used throughout the paper.

| | |
|---|---|
| $N$ | Number of agents. |
| $T$ | Number of time steps. |
| $K$ | Number of actions. |
| $W_t$ | Gossip matrix. |
| $\lambda_1, \lambda_2, \ldots, \lambda_N$ | Eigenvalues of $P$ sorted by norm, i.e. $|\lambda_1| > |\lambda_2| \geq |\lambda_3| \geq \cdots \geq |\lambda_N|$. It is always $\lambda_1 = 1 > |\lambda_2|$. |
| $\mu_{i,k}$ | global mean reward of arm $k$ on agent $i$. |
| $\mu_k$ | global mean reward of arm $k$. |
| $\Delta_i$ | Reward gaps, i.e. $\mu_1 - \mu_i$. |
| $p$ | the link probability $p$ of the Random graph |
| $\mathcal{G} = (\mathcal{V}, \mathcal{E})$ | base graph. |
| $\mathcal{G}_t = (\mathcal{V}, \mathcal{E}_t)$ | Communication network at time t. |
| $\mathcal{N}_i$ | the neighbors of agent $i$ in the base graph $\mathcal{G}$ |
| $\mathcal{N}_i(t)$ | the neighbors of agent $i$ in the communication network at time $t$ |
| $T_{i,k}(t)$ | Number of times arm $k$ is pulled by node $i$ up to time $t$. |
| $X_{i,k}(t)$ | Local reward of arm $k$ on agent $i$ at time step $t$. |
| $\widehat{\mu}_{i,k}(t)$ | Local estimate of $\mu_{i,k}$ on agent $i$ at time step $t$ . |
| $z_{i,k}(t)$ | Global estimate of $\mu_k$ on agent $i$ at time step $t$ . |
| $A_i(t)$ | Action played by agent $i$ at time $t$. |

