# OpenReview forum: "Distributed Multi-Agent Bandits Over Erdős-Rényi Random Networks"
_NeurIPS.cc/2025/Conference — NeurIPS 2025 poster_

### Official Review · Reviewer_PxL7 · 2025-06-29

**Clarity:** 3
**Significance:** 4
**Originality:** 3
**Rating:** 5
**Confidence:** 3

**Summary:**

The authors consider a random graph setting where the nodes are agents each of whom can select several arms for rewards. The stochasticity in the graph drives whether the agents communicate with each other based on the existence of a sampled edge. They then use gossip based reward estimation algorithm for agents to get both a personal and a global idea of the rewards that the arms provide and then which arms to keep and which to do away with. For this stochastic and challenging setting, they first upper bound the agent-wise and global community expected regret, then they interpret those results based on the obtained expression terms, ultimately relating it to graph parameters of p and graph connectivity (using Laplace matrix of the base graph). They discuss some specific upper boundds on known graph structures as examples. Then they proceed with lower bound definition on such class of problems. For these assumptions they have a LB estimate that is again a reinterpreted term from the UB. Then their experiments follow to validate their the bounds on the algorithm used on this problem.

**Questions:**

1. Its mentioned in the experimental settings that regret graphs show confidence intervals too. But the plots (Figure 2) doesnt clearly illustrate the same. Can this be made clearer?
2. Line 337 (...we sample each <> independently...) seems to have a missing term. Is it "edge"?
3. In line 338, what is this new term $q_i$ used to define means? They dont seem to have a definition either above or in the appendix table symbology! Is this expected?
4. In line 348 authors briefly discuss the algorithm's performance based on the rate at which agents discard their dominated arms. A graph on these dynamics, something like %arms discarded over total arms discarded along with the regret in a temporal phase, like in Figure 1 would be very informative, right? This could help clearly define for the agents' exploitation phase of farming rewards from the "trusted" arms!

**Ethical Concerns:**

["NO or VERY MINOR ethics concerns only"]

**Final Justification:**

The authors have answered my queries, including results from additional experiments and clarified regret formulation presentation for their revision. As acknowledged before, I'll keep my score.

**Limitations:**

1. A bit mild on the multi-agent side of problem formulation: getting information from agents, even ones you are neighbors with can be noisy/adversarial from the agents' perspective! Basically, assuming agents can always get a noiseless global estimate is a milder study on behalf of multi agent interaction. This enforces their "distributed" nature of their algorithm on a practical domain.
2. While authors already clearly acknowledge their algorithm suffers from a tradeoff between regret minimization and communication cost reduction, they could probably benefit from analyzing the communication in a broadcasted sense that emerges from the connected components in the $\mathcal{G}_t$, rather than a one-to-one communication between edges several times.

**Paper Formatting Concerns:**

no major concern here

**Quality:**

3

**Strengths And Weaknesses:**

Strengths:
1. Consideration of a practical yet less investigated problem in contemporary literature using random graphs with Multi-Arm Bandits for communication.
2. Work translates to stricter bounds on simplified problems already studied in the literature.
3. Actionable interpretation of the regret bound terms: global regret, $delta_k$, connectivity parameters $p$ and eigen values of the graph.
4. A sound initiative to define minimum requirements on the class of problem and prepare a lower bound on the same.
5. While the originality is not with the algorithm itself, its application to a defined problem and its performance bounds is novel.

Weakness:
1. A bit mild on the multi-agent side of problem formulation: getting information from agents, even ones you are neighbors with can be noisy/adversarial from the agents' perspective!
2. In section 3 lines 146-166, authors could be upfront about why they define both local and global reward definitions and how would it be helpful in the algorithm later! Specifically, line 151 (However...) could be made more concrete. Specifically in 173-175 they define the global objective that the policy should optimize. But what is individual agent trying to optimize? Local estimate of sampled arms? Global estiamte of sampled arms? A combination of them with a confidence bound? This only gets concrete by 201-204 when the actual algorithm details appear. An equivalent of equation (1) for individual agents can be made more explicit, just as they did so for the regret bounds later.

---

> ### Author Rebuttal · Authors · 2025-07-31
>
> We thank the reviewer for the valuable suggestions and address the concerns as follows.
>
> **Q1:** Regarding concern on the confidence interval in Figure 2
>
> **A1:** Thank you for your careful review. We would like to note that Figure 2 includes confidence intervals in a different format from Figure 1, using error bars rather than shaded regions (as in Figure 1). We chose this format because the confidence intervals are relatively small in this case, which may make them less visually prominent. This small variance across runs also supports the robustness and stability of our proposed algorithm, GSE. To improve clarity, we will revise Figure 2 either to be consistent with Figure 1 by using shaded areas or to enlarge the error bars to make the confidence intervals more apparent.
>
>
> **Q2:** Line 337 (...we sample $<>$ each  independently...) seems to have a missing term. Is it "edge"? In line 338 , what is this new term $q_i$ used to define means? They dont seem to have a definition either above or in the appendix table symbology! Is this expected?
>
> **A2:** Thank you for pointing this out. The missing term in Line 337 should be $q_i$. The variable $q_i$ is intentionally introduced to model the heterogeneity across agents by ensuring that each agent $i \in [N]$ has a different reward distribution for the same arm $k$. Specifically, $q_i$ is independently and uniformly sampled from $[0,1]$ at the beginning, and the local mean reward is defined as $\mu_{i,k} = q_i \cdot \frac{k-1}{K-1}$, which induces global heterogeneity. This experimental design follows a similar approach as in [1], and we will revise the text to clearly define $q_i$ and eliminate any ambiguity in the revised version.
>
>
>
> **Q3:** In line 348 authors briefly discuss the algorithm's performance based on the rate at which agents discard
> their dominated arms. A graph on these dynamics, something like that arms discarded over total arms discarded along with the regret in a temporal phase, like in Figure 1 would be very informative, right? This
> could help clearly define for the agents' exploitation phase of farming rewards from the "trusted" arms!
>
> **A3:** To further investigate the dynamic behavior of our algorithm during the arm elimination phase, we track the regret progression at each point where an arm is eliminated. Table 4 records five independent trials under a classical Erdős–Rényi communication graph with $p=0.9$. After averaging over 20 trials, we report the average timestep when an arm is eliminated and the average corresponding cumulative regret. We observe a consistent trend across experiments: each elimination event corresponds to a gradual transition from exploration to exploitation, as reflected in the slower growth rate of regret after the elimination of more suboptimal arms. This illustrates that agents progressively narrow their action set to focus on near-optimal arms, thereby stabilizing their regret growth. This temporal analysis provides empirical support for our theoretical discussion, and highlights the importance of timely arm elimination in reducing long-term regret.
>
>
> *Table 4: Average over 20 runs of GSE with classical Erdős–Rényi graph and p=0.9.*
>
> | # Eliminated Arms | Avg. Timestep | Avg. Regret |
> |-------------------|--------------:|------------:|
> | 1                 |        332.0  |     81.9875 |
> | 2                 |        372.4  |     89.5367 |
> | 3                 |        443.4  |     98.1758 |
> | 4                 |        650.2  |    111.0484 |
>
>
> **Q4:** A bit mild on the multi-agent side of problem formulation: getting information from agents, even ones you are neighbors with can be noisy/adversarial from the agents' perspective!
>
> **A4:** We appreciate the valuable suggestions and would include the discussion on scenarios with noisy and adversarial agents in the future work section as well. The discussion reads as follows verbatim. 1. Noisy agents: When information from other agents is noisy—meaning that the shared reward data consists of the original reward signal plus potentially random noise or has correlation risks [4], as commonly observed in digital communication channels—this issue can, in principle, be addressed using techniques from noisy optimization [6]. These methods combine statistical and optimization tools to effectively handle function optimization under noise, which can be leveraged in the construction of reward estimators when using such noisy information. Also, this problem is novel and meaningful in the sense that random communication and random noise coexist, making the noisy optimization problem more challenging. 2. Adversarial agents: When other agents are adversarial—intentionally broadcasting malicious information—we will adopt a Byzantine-robust framework [2, 5], in which the reward information is truncated to produce robust estimators, such as the trimmed mean or median of means, to filter out malicious reward information. Notably, this direction is novel in that the presence of random graphs with underlying base graphs further complicates the construction of Byzantine-robust estimators, compared to traditional multi-agent bandits or distributed robust optimization settings with fixed, connected graphs or assuming linear reward models.
>
>
> **Q5:** While authors already clearly acknowledge their algorithm suffers from a tradeoff between regret minimization and communication cost reduction, they could probably benefit from analyzing the communication in a broadcasted sense that emerges from the connected components in the $\mathcal{G}_t$, rather than a one-to-one communication between edges several times.
>
> **A5:** We appreciate this suggestion and will include the idea of communication cost reduction through the information propagation of graph-level analysis rather than edge-level analysis in the discussion of future work. Hypothetically, communication based on connected components can be illustrated using $l$-connected graphs [3], which consider the composition of subgraphs across time steps. Instead of analyzing a single graph at each time step, we consider the joint information provided by graphs from previous steps, which could potentially further reduce communication costs.
>
> [1] Jialin Yi and Milan Vojnovic. Doubly adversarial federated bandits. In the International Conference on Machine Learning, 2023.
>
> [2] Vial, Daniel, Sanjay Shakkottai, and R. Srikant. Robust multi-agent bandits over undirected graphs. Proceedings of the ACM on Measurement and Analysis of Computing Systems, 2022
>
> [3] Zhu, Jingxuan, and Ji Liu. Distributed multiarmed bandits.  IEEE Transactions on Automatic Control, 2023
>
> [4] Shao, Qi, Jiancheng Ye, and John CS Lui. Risk-aware multi-agent multi-armed bandits. Proceedings of the Twenty-fifth International Symposium on Theory, Algorithmic Foundations, and Protocol Design for Mobile Networks and Mobile Computing, 2024.
>
> [5] Mitra, Aritra, Arman Adibi, George J. Pappas, and Hamed Hassani. Collaborative linear bandits with adversarial agents: Near-optimal regret bounds. Advances in neural information processing systems, 2022
>
> [6] Mai, Van Sy, Richard J. La, Tao Zhang, and Abdella Battou. Distributed optimization with global constraints using noisy measurements. IEEE Transactions on Automatic Control, 2023

---

> > ### Comment · Reviewer_PxL7 · 2025-08-04
> >
> > I thank the authors in addressing my queries, most of which are satisfactorily answered in actionable minor edits on their part. I have a few followups,
> >
> > 1. As per the original review, does the section Weakness point (2), entail some planned refactor from the authors’ part?

---

> > > ### Author Response · Authors · 2025-08-05
> > >
> > > We appreciate the reviewer’s very detailed comments, for reading the responses, and for this follow‑up question. We apologize that we thought the weakness part was the same as the limitation part by looking at the first weakness and the first limitation, which coincides with one another, and thus we missed the second weakness part by treating it the same as the second limitation. We sincerely appreciate your patience with us and for pointing out this question. Our response to the question and the proposed modifications are as follows:
> > >
> > > **Response to the motivation/reason for defining the local and global reward, and the objective of local agents**
> > >
> > > The definition of the local reward $X_{i,k}(t)$ is the same as in the single‑agent stochastic bandit, which allows us to define the local reward mean $\mu_{i,k}$ by taking the expectation of $X_{i,k}(t)$. The global reward $X_{k}(t) = \frac{1}{N}\sum_{j\in[N]} X_{j,k}(t)$ is defined as the average of the local rewards over all agents, which is a meaningful consensus quantity. By applying the linearity of expectation, we define the global reward mean as $\mu_k = \frac{1}{N}\sum_{j\in[N]} \mu_{j,k}$, which is the average of the local reward means and is the shared target of all agents across all time steps. Notably, both the local and global reward mean, $\mu_{i,k}$ and $\mu_k$, are unknown, necessitating inference and information aggregation across agents. Clearly defining this time‑invariant shared target helps formulate the cooperative multi‑agent bandit problem in the language of regret minimization in a stochastic setting and makes the algorithm design meaningful.
> > >
> > > Objective of individual agent: Notably, the local agent's objective is to infer the global optimal arm by estimating the global reward mean for each arm, formally captured by minimizing individual regret (defined below). The uncertainty in these reward estimations necessitates incorporating a confidence bound in the final estimation used in the decision criterion in Eq. (4). We initially considered including the definition of individual regret (objective) in the problem formulation but removed it due to space limitations. However, as raised by the reviewer, this might have affected the clarity of the presentation. Thanks for your suggestion which helps us better communicate the setting.
> > >
> > > **Modifications: Accordingly, we will modify the expressions mentioned in the reviews as follows, verbatim.**
> > >
> > > Line 150-153: When agent $i$ selects arm $A_i(t)$ at round $t$, it observes only a local stochastic reward $X_{i,A_i(t)}(t)$ drawn independently from the distribution $\mathbb{P}\_{i,A\_i(t)}$ each round, reflecting the real‑world scenario where each agent observes a local reward from a time‑invariant distribution per round. However, the agent's true objective depends on the global reward $X\_{A\_i(t)}(t)$, where $X\_{A\_i(t)}(t)\coloneqq \frac{1}{N}\sum\_{j\in[N]} X\_{j,A\_i(t)}(t)$, which is the average reward over all agents and is not observable by any agent.
> > > Accordingly, we define the global mean reward for arm $k$ as $\mu_k\coloneqq \frac{1}{N}\sum_{j\in[N]} \mu_{j,k}$ by taking the expectation of $X_{A_i(t)}(t)$, where $\mu_{j,k}$ is the mean of $\mathbb{P}_{j,k}$. The underlying target is to optimize the global reward, and hence agents require estimates of the global reward mean values for each arm in order to identify the global optimal arm with the highest global mean reward. With this objective, we aim to design algorithms that allow agents to minimize regret with respect to the global optimal arm.
> > >
> > > Line 173-175: The global objective, as well as the objective for each local agent, is to design a distributed algorithm $\pi$ that minimizes the total global regret over $T$ rounds; in other words, all agents share a common target that makes consensus possible. To make this coherent, we first define the individual regret, which represents the objective of local agent $i \in [N]$, as:$$\text{Reg}\_{i,T}(\pi)=T\mu^\*-\sum_{t=1}^T\mu_{A_i(t)}(t)=\sum_{t=1}^T\Delta_{A_i(t)},$$where $\mu^\*=\max_{k\in[K]}\mu_k$ is the optimal global average reward, and $\Delta_k=\mu^{*}-\mu_k$ is the global suboptimality gap for arm $k$.
> > >
> > > Notably, local agents aim to optimize the global reward means of the arms they pull, i.e., to pull the global optimal arm. We define the local regret as the cumulative difference between the global reward mean of the global optimal arm and that of the actual arm pulled by each local agent. Building on this, we define the global regret of the entire system for algorithm $\pi$ as the sum of the individual regrets over all agents:$$\text{Reg}\_T(\pi)=\sum\_{i\in[N]}\text{Reg}\_{i,T}(\pi)=NT\mu^\*-\sum_{i\in[N]}\sum_{t=1}^T\mu_{A_i(t)}=\sum_{i\in[N]}\sum_{t=1}^T\Delta_{A_i(t)},$$which establishes the equivalence of Eq.(1) in the context of individual agents.
> > >
> > > We have read through all the comments and ensure that all comments are addressed herein. We truly appreciate your valuable comments.

---

> > > > ### Comment · Reviewer_PxL7 · 2025-08-05
> > > >
> > > > I thank the authors in addressing the comments promptly. I have also reviewed discussions from other review threads, especially pertaining to the core of their study: incremental novelty in both algorithmic and theoretical bounds, in the formulated setting. Conditioned on the proposed minor changes and edits, I’ll keep my score.

---

> > > > > ### Author Response · Authors · 2025-08-05
> > > > >
> > > > > We sincerely thank the reviewer for carefully reading our paper and responses, providing valuable advice, and reviewing the discussions with other reviewers as well. These have been very helpful and insightful. We will ensure that the discussed points are incorporated in the revised paper. Specifically, we will: 1) revise Figure 2 to be consistent with Figure 1 and to clearly indicate the confidence intervals, 2) clarify $q_i$ in the experimental settings, 3) add further experiments and figures visualizing the table in the response about the eliminated arms, either in the main body or in the appendix, 4) revise the writing from Lines 146–166 and 173–175 to further clarify the definitions and motivations of local and global reward estimates as well as the objective of the local agent as described in the reply, and 5) add a discussion about noisy/adversarial agents and communication reduction to the future work section.
> > > > > Again, we sincerely appreciate the reviewer's detailed review, the recognition of our work, and the positive attitude.

---

### Official Review · Reviewer_6FB3 · 2025-07-01

**Clarity:** 3
**Significance:** 2
**Originality:** 2
**Rating:** 5
**Confidence:** 4

**Summary:**

This paper studies a distributed multi-agent multi-armed bandit (MA-MAB) problem in a heterogeneous reward setting where agents communicate over time-varying random graphs derived from a fixed base graph via the Erdős–Rényi model. The authors propose a fully distributed algorithm that combines successive arm elimination with a gossip-based communication protocol, enabling agents to estimate global rewards despite potentially disconnected communication at each round. Theoretical analysis shows that the algorithm achieves a regret bound of order $O(\log T)$, which is highly interpretable in terms of the graph properties and connectivity. The paper also establishes a matching lower bound up to constant factors, and provides empirical results validating the regret–communication trade-off and superiority over prior work.

**Questions:**

As noted earlier, my primary concerns relate to the level of novelty and the tightness of the proposed regret bounds.
1. Could you clarify the main technical challenges in establishing the regret upper bounds in Lemma 5.2 and Theorem 5.3? In particular, what are the key obstacles in reducing the second and third terms in Theorem 5.3? Do you believe it is possible to improve these terms to at least match known bounds in the time-invariant, homogeneous setting [1]?
2. Additionally, in Theorem 5.9, what distinguishes your proof of the regret lower bound from the arguments of a similar lower bound for the single-agent case in [1], which your approach seems to closely follow? Is there potential to strengthen the lower bound by incorporating graph-dependent terms, such as the communication probability $p$?
3. For a revised version, I recommend moving the Conclusion and Future Work section into the main text, while optionally relocating part of the Related Work section to the appendix if space is limited.

[1] Martínez-Rubio, David, Varun Kanade, and Patrick Rebeschini. "Decentralized cooperative stochastic bandits." Advances in Neural Information Processing Systems 32 (2019).\
[2] Lattimore, Tor, and Csaba Szepesvári. Bandit algorithms. Cambridge University Press, 2020.

**Ethical Concerns:**

["NO or VERY MINOR ethics concerns only"]

**Final Justification:**

My primary concerns were with the level of novelty in this work, and with the gaps between upper and lower bounds, and between the heterogeneous and homogeneous settings. These concerns have been partially addressed by a well-written rebuttal, which makes me more comfortable recommending this work for publication. Please see my response to the rebuttal below for more details.

**Limitations:**

Yes

**Paper Formatting Concerns:**

I did not notice any major issues.

**Quality:**

2

**Strengths And Weaknesses:**

**Strengths:**
- **Quality:** The paper’s main claims are supported by rigorous and well-structured proofs, demonstrating a solid theoretical foundation. Additionally, the paper includes a basic experimental evaluation that demonstrates the superiority of the proposed method.
- **Clarity:** The proposed algorithm is well-motivated and clearly explained, with intuitive justifications provided in the main text. The authors carefully analyze the theoretical results, offering meaningful comparisons with prior work. Finally, the effectiveness of the method is supported by a set of illustrative experimental evaluations.
- **Significance:** This work advances the state-of-the-art by providing an algorithm with provable regret bounds that operates in a distributed and heterogeneous environment with more limited connectivity than considered in prior works [1,2]. Additionally, the regret bounds improve the state-of-the-art in a relaxed setting [1].
- **Originality:** Although the authors build on established techniques such as running-consensus-based estimation [3] and successive elimination-based implicit synchronization [4], they employ them in a novel manner in the random connectivity heterogeneous setting to achieve an algorithm with improved performance.

**Weaknesses:**
Rather than separating into broad quality, clarity, significance and originality categories, I will outline my main concerns in a more detailed manner below.
- My main concern lies in the tightness of the proposed regret bounds. While the first term in the upper bound of Theorem 5.3 matches the lower bound and scales as $O(\sum \log T /\Delta_k)$, the remaining terms—particularly $O(KN^2\log T/p)$—dominate the overall regret (up to the reward gaps). This discrepancy is not adequately addressed in the proposed lower bound or in the accompanying discussion in Remark 5.10, leaving a notable gap between the upper and lower bounds. Furthermore, it appears that even in homogeneous settings with time-invariant graphs, the regret of the proposed algorithm still scales as $O(KN^2\log T)$, which is significantly worse than the bounds achieved in prior work, such as [3]. That said, this limitation seems to be common in the literature, and as acknowledged in Remark 5.5, the authors’ results nonetheless represent a meaningful improvement over those in [1,2].
- Given the aforementioned limitations, this work represents an incremental advancement in the state of the art for heterogeneous MAMAB under time-varying communication graphs. While it offers meaningful improvements, several known challenges in this setting remain unresolved.
- The techniques employed in this work—such as successive elimination [4], running consensus estimation [3], and the lower bound proof strategy [5]—are based on established methods, which limits the overall novelty of the contribution.

Other issues –
- Implementing the proposed algorithm requires agents to have global knowledge of certain properties of the communication graph, such as the eigenvalues of its Laplacian. Although this assumption has been adopted in prior work [3], it remains unrealistic in many practical settings.
- Although the edge communication probability $p$ can be estimated during a burn-in phase, this introduces an additional term in the regret upper bound—$O(N^2\log T/p^2)$ (Theorem D.2)—which can be larger and further impacts the algorithm’s performance.
- In the proposed algorithm, agents exchange their estimates of the global mean for each arm in every round, resulting in high communication overhead—linear in $T$—and complex message structures. Unfortunately, the communication cost is neither optimized nor analyzed at the bit level in this work.
- The main text does not include proof outlines for Lemma 5.2 and Theorem 5.3. Given that this is primarily a theoretical contribution, providing such outlines is important for guiding readers through the core ideas of the analysis. I strongly recommend including them in a revised version of the paper.
- The authors claim that the numerical experiments validate the theoretical regret scaling with problem complexity; however, Figure 2 only illustrates the general trend of regret as the communication probability $p$ and base graph connectivity vary. It does not empirically validate the precise dependence on these factors as specified in the regret upper bound of Theorem 5.3, which I suspect may be loose.
- Additional proofreading would improve the paper. For example, in line 81 “the” should be removed, and the sentence in lines 190-191 has been stated before in the same paragraph.


[1] Xu, Mengfan, and Diego Klabjan. "Decentralized randomly distributed multi-agent multi-armed bandit with heterogeneous rewards." Advances in Neural Information Processing Systems 36 (2023): 74799-74855.\
[2] Zhu, Zhaowei, et al. "Federated bandit: A gossiping approach." Proceedings of the ACM on Measurement and Analysis of Computing Systems 5.1 (2021): 1-29.\
[3] Martínez-Rubio, David, Varun Kanade, and Patrick Rebeschini. "Decentralized cooperative stochastic bandits." Advances in Neural Information Processing Systems 32 (2019).\
[4] Yang, Lin, et al. "Cooperative multi-agent bandits: Distributed algorithms with optimal individual regret and constant communication costs." arXiv preprint arXiv:2308.04314 (2023).\
[5] Lattimore, Tor, and Csaba Szepesvári. Bandit algorithms. Cambridge University Press, 2020.

---

> ### Author Rebuttal · Authors · 2025-07-31
>
> Thanks for your positive feedback. We address your concerns as follows.
>
> **Q1:** Concerns about algorithm novelty
>
> **A1:** Reviewer nBgW raised a similar concern; please see Q3 and A3 in response to reviewer nBgW for a detailed explanation of algorithmic novelties and corresponding theoretical analysis.
>
> **Q2:** Concerns about knowledge of $\lambda_{N-1}$
>
> **A2:** We appreciate the reviewer’s concern about algebraic connectivity $\lambda_{N-1}$. Similar to prior work [2,3], our confidence bound requires knowledge of $\lambda_{N-1}$. However, $\lambda_{N-1}$ can be estimated in a fully decentralized manner (e.g. brief gossip-based power iteration [7]).
>
> **Q3:** Concerns about additional term on estimating $p$
>
> **A3:** We appreciate and agree that estimating $p$ without prior knowledge inevitably incurs extra regret. In future work, we plan to reduce this overhead by (1) improving the sample efficiency of the estimation phase, or (2) incorporating the estimated $p$ into decision making during exploitation instead of requiring the exact value.
>
> **Q4:** Concerns about the communication complexity
>
> **A4:** Thanks for highlighting this point. As discussed in Remark 5.6, the average communication complexity per agent in our setting is $O(p|\mathcal{E}|T/N)$, where $p$ is the edge activation probability and $\mathcal{E}$ is the edge set of the base graph with $|\mathcal{E}| \leq O(N^2)$. In contrast, [1] assumes a complete graph and incurs $O(pNT)$ communication. Notably, when $|\mathcal{E}|=O(N)$, our complexity becomes $O(pT)$, independent of $N$, indicating better scalability. Exploring ways to further reduce communication—building on recent advances in cost-efficient protocols [5]—is an exciting future direction.
>
> **Q5:** Challenges in upper bounds in Lemma 5.2 and Theorem 5.3
>
> **A5:** Thanks for the question. The main technical challenge in establishing the regret bounds in Lemma 5.2 and Theorem 5.3 lies in the general communication model we consider. Unlike prior works that assume fixed or strongly connected graphs w.h.p., our setting allows each edge to appear independently with probability $p>0$, meaning the graph may be disconnected in many rounds. As a result, traditional confidence bounds—designed to handle sampling noise—are no longer sufficient, since error also arises from communication delays. The key challenge is thus to design a confidence interval that accounts for both statistical estimation and consensus error.
>
> To this end, we introduce the confidence interval $c_{i,k}(t)$ in Eq.(4), which captures both estimation and consensus error. The regret bounds in Lemma 5.2 and Theorem 5.3 build upon Lemma 5.1, which guarantees convergence of each agent’s estimate $z_{i,k}(t)$ to the global mean $\mu_k$ under this interval. The proof involves defining several high-probability events (Appendix C, Line 520) ensuring: (i) concentration of local estimates, (ii) sufficient mixing of information across agents to reach consensus, and (iii) balanced arm pulls for different agents. These allow us to decompose the error in $z_{i,k}(t)$ into estimation and consensus components (Eq.(12)), which are then bounded separately. We will incorporate the above explanation into the revised version of the paper.
>
> **Q6:** Tightness of regret bounds and obstacles in reducing upper bound
>
> **A6:** We acknowledge that there remains a gap between the theoretical upper and lower bounds in our current analysis. Specifically, while the first term in Theorem 5.3—which captures the centralized noise matches the lower bound in Theorem 5.9, the remaining terms in the upper bound depending on graph properties such as $p$ and $\lambda_{N-1}$, are not yet matched by our lower bound. However, we would also like to add that we make contributions toward narrowing the gap between the lower and upper bounds in existing work. Notably, our upper bound is smaller by the order of $O(N)$ compared to that in [3] for the centralized regret term under their assumptions. More importantly, it also introduces a novel reflection of the communication graph through the presence of $1/p$ in the upper bound. Our newly added experiment (A3(ii), Table 2 in responses to reviewer bG2e) shows that the actual regret exhibits a dependency of $1/p^{0.93}$, which numerically validates the near-tightness of our upper bound. On the other hand, our lower bound improves upon existing work [1] by explicitly incorporating the sub-optimality gap.
>
> Moreover, we believe that the $O(N^2/p)$ term in the upper bound reflects a fundamental limitation due to communication delays and is unlikely to be avoided. Consider, for instance, a communication graph structured as a cycle, where each edge is connected independently with probability $p$ in each round. In such a scenario, it takes on average $O(N/p)$ rounds for information from one agent to reach all other agents (similar philosophy as suggested by the reward delay in [6]). Since each agent must collect information from all others to compute a globally optimal policy, this implies a minimum communication delay of $O(N/p)$ for each agent. Summing up across all $N$ agents, this suggests a lower bound of $O(N^2/p)$ on the global regret, indicating that the communication-induced term in Theorem 5.3 is intrinsic to the problem. While we have explored ideas for reducing the last two terms in Theorem 5.3 (e.g., potentially improving dependency on $K$ in the third term from $O(K)$ to $O(\log K)$, as discussed in our reply to Reviewer bG2e), we still anticipate that any meaningful upper bound must involve a term of at least $O(N^2/p)$ to account for the global information aggregation delay. Closing this remaining gap and formally establishing a lower bound that includes $p$ and $\lambda_{N-1}$ remains an important direction for future work.
>
> **Q7:** Possibility to improve the graph-dependent terms to match bounds in time-invariant, homogeneous setting
>
> **A7:** While there remains a gap between our regret bound in the time-invariant and homogeneous setting and the result in [2], we believe that this gap is not merely due to our analysis, but rather stems from a fundamental difference between the homogeneous and heterogeneous settings which results in the difference in the algorithm design.
>
> The dependence on $N$ in our second and third terms is $O(N^2)$, while the corresponding dependence in Theorem 3.2 of [2] is only $O(N\log N)$. As explained in A6, in the heterogeneous setting, each agent must collect information from *all* other agents to accurately estimate the global reward and make optimal decisions. This necessity leads to an inherent $O(N^2)$ communication delay (even in the time-invariant case when $p=1$). In contrast, in the homogeneous setting, agents share the same reward structure and do not necessarily need information from others, enabling a regret bound with only $O(N\log N)$ dependence. This core difference fundamentally limits the possibility of matching the bounds in [2] within our heterogeneous setting.
>
> Given this, we see potential to improve our bounds via algorithmic modifications for best-of-both-worlds performance—i.e., optimality in both heterogeneous and homogeneous settings. One idea is to detect reward consistency across agents and switch accordingly between heterogeneous and homogeneous algorithms, akin to prior work adapting to stochastic vs. adversarial regimes [4]. This may also reduce the $\log T$ dependence in Theorem 5.3, as achieved in [2] for homogeneous setting. We leave this for future work.
>
> **Q8:** Distinctions between our lower bound and single-agent lower bound and the possibility to add graph-dependent terms in our lower bound
>
> **A8:**  Our proof extends beyond the single-agent case by directly targeting *global* regret across all $N$ agents. Unlike classical lower bounds that perturb one arm's mean for a single agent, we construct two instances $\nu$ and $\nu'$ that shift the mean of the same arm for all agents simultaneously (see lines 305–308), yielding a lower bound on $\sum_{i=1}^N \mathbb{E}[T_{i,k}(T)]$ and thus the global regret. This primarily establishes the tightness of the centralized term. That said, we acknowledge that the lower bound is not our main contribution. The key novelties lie in: 1) a *new setting* with generalized random graphs; 2) a *new algorithm* based on successive elimination with a novel confidence bound adapted to random gossip; and 3) an *upper-bound analysis* that explicitly captures graph parameters ($p$, $\lambda_{N-1}$), removing the strong assumptions on $p$ in [3]. Still, our lower bound improves upon [1], and we agree that extending it to incorporate graph-dependent terms remains a challenging and important direction.
>
> **Q9:** Regarding the concerns about the placement of conclusion, future work and proofreading modifications
>
> **A9:** Thank you for the suggestions. We will move the conclusion and future work to the main body, proofread, and incorporate these changes in the revision.
>
> References
>
> [1] Mengfan Xu and Diego Klabjan. Multi-agent multi-armed bandit regret complexity and optimality. AISTATS, 2025.
>
> [2] David Martínez-Rubio, Varun Kanade, and Patrick Rebeschini. Decentralized cooperative stochastic bandits. NeurIPS, 2019.
>
> [3] Mengfan Xu and Diego Klabjan. Decentralized randomly distributed multi-agent multi-armed bandit with heterogeneous rewards. NeurIPS, 2023.
>
> [4] Bubeck, Sébastien, and Aleksandrs Slivkins. The best of both worlds: Stochastic and adversarial bandits. COLT, 2012.
>
> [5] Xuchuang Wang, Lin Yang, Yu-Zhen Janice Chen, Xutong Liu, Mohammad Hajiesmaili, Don Towsley, John C.S. Lui. Achieving near-optimal individual regret low communications in multi-agent bandits. ICLR, 2023.
>
> [6] Tal Lancewicki, Shahar Segal, Tomer Koren, Yishay Mansour. Stochastic multi-armed bandits with unrestricted delay distributions. ICML, 2021.
>
> [7] David Kempe, Alin Dobra, and Johannes Gehrke. Gossip‐based computation of aggregate information. FOCS, 2003.

---

> > ### Comment · Reviewer_6FB3 · 2025-08-06
> >
> > I thank the authors for their detailed and well-written rebuttal.
> >
> > Regarding the novelty concern, the authors argue that their algorithm is carefully designed to address the time-varying heterogeneous setting, incorporating a new confidence interval and a weight matrix. While I maintain that these techniques are not particularly novel relative to prior works, I acknowledge that the overall contribution is sufficient. In particular, the trade-offs captured in the second term of the regret upper bound are of interest to the community.
> >
> > My remaining primary concerns relate to the gap between the upper and lower bounds, as well as the suboptimal performance of the algorithm in homogeneous environments. The authors suggest that some of the upper bound terms—such as the $O(N^2/p)$ term—may be unavoidable due to the heterogeneous nature of the problem. While a formal justification for this claim would have been preferable, and the presence of a $O(\log T)$ term remains unaddressed, the authors' analysis, combined with demonstrable improvements over existing bounds, makes me more comfortable recommending this work.
> >
> > In light of the above points, and after considering the rebuttal and discussions with other reviewers, I have decided to raise my score. I encourage the authors to incorporate these discussions into the revised version, particularly those concerning the gaps between lower and upper bounds, and between the heterogeneous and homogeneous settings.

---

> > > ### Author Response · Authors · 2025-08-07
> > >
> > > We sincerely appreciate your valuable suggestions and recognizing the importance of our work. We are also thankful for your acknowledgment of our overall contributions, particularly the relevance of the trade-offs between regret and communication, as well as the improvements over existing bounds achieved through our novel analysis.
> > >
> > > We will add the discussion on the improvement over the existing bounds as a remark in the revised paper. Specifically, we will include the above discussion on the tightness of the result,  and the improvement of the lower bound and upper bound, and communication cost over existing results in the analysis section as two remarks (after the lower bound proposed in Section 5.2).
> > >
> > >
> > > For the concern about the gap between the upper and lower bounds, we will add a discussion on the necessity of the $O(N^2/p)$ term in the lower bound in the revised version, and formally state that establishing this necessity via a lower bound theorem involving $O(N^2/p)$ is an interesting direction for future work. Related to the presence of the $\log T$ term in the last two terms of our upper bound, this term arises from the high-probability guarantee of the good event defined in our analysis (line 520), which ensures that reward information from any agent can be passed to other agents within a limited number of rounds with high probability. It is also worth investigating whether this $\log T$ dependency in the last two terms can be avoided using new tools, similar to the case of the homogeneous MA-MAB problem with fixed communication graphs (where $p = 1$) studied in [2]. We will add this point to the future work section verbatim.
> > >
> > >
> > > Additionally, for the remaining points suggested in your comments, we will ensure to incorporate them in the revised version. Precisely, we will 1) introduce our algorithmic novelty as well as theoretical analysis novelty in more detail in Sections 4 and 5; 2) discuss the limitations regarding the knowledge of $\lambda_{N-1}$ and the estimation of $p$ in the conclusion and future work section; 3) discuss the communication complexity advantage of our work compared to previous studies after Remark 5.6, and introduce reducing this complexity as a future direction in the conclusion; 4) clarify the challenges in obtaining the upper bounds in Lemma 5.2 and Theorem 5.3 by expanding the proof sketches in Section 5; 5) add a discussion on the tightness of the upper and lower bounds, along with a comparison with the results in [2]; 6) move the conclusion and future work to the main body; 7) perform another round of proofreading to further polish the paper, e.g. line 426; 8) include the additional experiments that demonstrate the numerical dependency on $p$ and $\lambda$ based on the rebuttal (Table 2 \& 3 in A3 for reviewer bG2e—we apologize for not including this in the previous rebuttal due to the word limit).
> > >
> > > Once again, we sincerely appreciate the reviewer’s detailed review, careful examination of our replies, and reconsideration of the score. The suggestions and discussions have been crucial in helping us further polish this work and inspiring future directions.

---

### Official Review · Reviewer_nBgW · 2025-07-02

**Clarity:** 3
**Significance:** 2
**Originality:** 2
**Rating:** 4
**Confidence:** 3

**Summary:**

This paper studies the problem of multi-agent multi-armed bandit. In particular, it accounts for a scenario where the rewards are heterogeneous across the $N$ agents and the communication graph is stochastic and sampled from an Erdos-Renyi model at every round.  The paper's main contribution is analyzing this setting, which generalizes the models that have been proposed in the literature so far: the authors propose an algorithm that achieves no-regret and highlight the components of the regret upper bound, that are interpretable; moreover, they provide a lower bound that proves that at least one component of the upper bound is tight. Finally, some experimental evaluations on both synthetic and real-world data are provided.

**Questions:**

Can the authors argue whether the upper bound is tight or not about the problem-dependent quantities?

This is a major point; if I missed some parts of the submission, and the authors can answer this positively, I would reconsider my score positively. However, note that this is not my only evaluation criterion, because I also think that the methodological contribution is not a breakthrough if compared to previous works in this literature.

**Ethical Concerns:**

["NO or VERY MINOR ethics concerns only"]

**Final Justification:**

The authors' arguments about the actual existence of a trade-off in the regret lower bound convinced me. This paper is not a breakthrough, but there are some contributions which makes it pend slightly more towards accept rather than reject. I'm fine with either decision anyways.

**Limitations:**

yes

**Paper Formatting Concerns:**

--

**Quality:**

2

**Strengths And Weaknesses:**

Strengths:

- The paper is overall well written, the narration can be followed without any trouble, and the message of the paper is clear.
- The comparison with existing results is fair and highlights the additional difficulties of this setting.
- The experimental campaign goes beyond the simple comparison with the competing algorithms and explores the links between the regret and the problem parameters.

Weaknesses:

- I don't like how the conclusions and future work are presented as an appendix. There are two cases: either this section is important (and I believe it is, as the future work is one of the most interesting parts of a paper) and you find a spot for that in the main paper; or this section is not crucial for the paper and you just avoid it. In both cases, having an appendix for this makes no sense in my opinion. This is not a major point, but I wanted to report it.

- The regret-communication trade-off that appears in the regret upper bound does not appear in the lower bound, and thus there is no way to be sure that this trade-off actually exists and is not a byproduct of the regret analysis. **This is a major point**: the narration of the paper strongly depends on this "instance-dependent" type of guarantee, and the meaning of the problem parameters appearing in it. The authors provide a lower bound, which basically resorts to the standard MAB lower bound but does not explore the presence of $p$ or other graph-dependent quantities. There is some algorithmic contribution, but since the overall approach is based on existing families of algorithms, the technical contribution is what can make this paper accepted or not, in my opinion.

- The methodological contribution is not a breakthrough if compared to previous works in this literature. The authors provide algorithmic routines to deal with the stochasticity of the graph and the heterogeneity of rewards simultaneously, but overall the main underlying ideas to tackle the MA-MAB problem have already been outlined in previous works.

---

> ### Author Rebuttal · Authors · 2025-07-31
>
> **Q1:** Concerns about conclusion and future works.
>
> **A1:** Thank you for the suggestion. We will revise the paper by moving the conclusion and future work to the final section of the main body.
>
> **Q2:** Concerns about tightness of upper bound and regret-communication trade-off existence
>
> **A2:** We acknowledge that verifying the presence of graph-dependent terms in the lower bound is essential to substantiate the regret-communication trade-off emphasized in our upper bound analysis. While Theorem 5.3 presents an upper bound that includes terms dependent on graph properties such as the link probability $p$, the lower bound in Theorem 5.9 does not yet capture these dependencies. However, we argue that the trade-off is inherent to the problem and not merely an artifact of the analysis, and we make contributions towards this direction.
>
> To support this claim, consider a cycle graph over $N$ agents, where each edge is activated independently with probability $p$. On average, it takes $O(N/p)$ rounds for information to propagate from one agent to reach all the other agents. Since our setting requires each agent to optimize with respect to global rewards, which depend on data from all agents, this information delay directly impacts the agent's ability to make optimal decisions. Therefore, each agent may suffer at least $O(N/p)$ regret, leading to a total global regret of at least $O(N^2/p)$. This suggests a fundamental dependence on $p$, even in the lower bound. Combined with the lower bound on the statistical noise term in Theorem 5.9, this observation highlights that the regret-communication trade-off should indeed exist. Intuitively, as $p$ decreases, it takes longer for information to diffuse across the network, resulting in larger regret. In the extreme case where $p$ approaches zero, inter-agent communication becomes virtually impossible, and no agent can optimize global performance, leading to linear regret. This aligns with the trade-off discussed in Remark 5.6: more frequent communication reduces regret, while sparse communication increases it.
>
> We would also like to add that we make contributions toward narrowing the gap between the lower and upper bounds in existing work. Notably, our upper bound is smaller compared to [1]. More importantly, it also introduces a novel reflection of the communication graph through the presence of $\frac{1}{p}$ in the upper bound. Surprisingly, our newly added experiment ((ii), Table 2 in A3 for Q3 raised by reviewer bG2e) shows that the actual regret exhibits a dependency of $\frac{1}{p^{1}}$, which numerically validates the near-tightness of our upper bound. On the other hand, our lower bound improves upon existing work [4] by explicitly incorporating the sub-optimality gap.
>
> We do sincerely agree that formally incorporating this communication-related dependency (e.g., via $p$ or $\lambda_{N-1}(\text{Lap}(\mathcal{G}))$) into the lower bound remains an important open direction. We hope our work could inspire future work on how to close the gap between upper and lower bounds in terms of graph-dependent terms.
>
> **Q3:** Concerns about our methodological contribution
>
> **A3:** The core contribution of our work lies in studying the multi-agent bandit problem under a general random communication graph, where each edge is independently active with probability $p\in(0,1]$. In contrast, prior works either assume strongly connected communication graphs at all times [2,3], or require the graph to be strongly connected with high probability at each round [1]. While [1] is the only work that explicitly incorporates the edge activation probability $p\ge 1/2+1/2\sqrt{1-(\epsilon/{NT})^{2/(N-1)}}$,
> which requires $p$ to be significantly large—approaching $1$ as $N$ or $T$ grows—and yields a regret bound ($O\left(\sum_{k:\Delta_k>0}\Delta_k^{-1}\log(T)+KN\log(T)\right)$ in [1]) that is worse than ours ($O\left(\sum_{k:\Delta_k>0}\Delta_k^{-1}N\log(T)\right)$ in ours) by a factor of $N$.
>
> In our novel and more general setting, our proposed algorithm (GSE) introduces several key methodological innovations compared to prior work. These include a new algorithmic framework based on arm elimination, refined weight matrix for gossip, and a carefully designed confidence interval that accounts for both estimation and consensus errors. Each of these components is tailored to the challenges of heterogeneous MA-MAB with random communication and plays a critical role in achieving stronger theoretical guarantees under minimal connectivity assumptions.
>
> **Algorithmically, we introduce them as follows:**
>
> (i) Algorithm framework: From the perspective of algorithmic protocol, our method adopts a *arm elimination* framework, whereas prior works in this literature primarily rely on *UCB-style algorithms* [1,2,3]. While arm elimination is not new, it is particularly well-suited to the heterogeneous MA-MAB setting. In this setting, each agent only observes local rewards, while the goal is to minimize cumulative *global regret*. To ensure the global estimate—obtained via gossip averaging—is unbiased and not overly influenced by individual agents’ feedback, it is important that all agents pull *each arm a comparable number of times*. Arm elimination naturally enforces such balance, as shown in Lemma B.4, which bounds pull count differences across agents. In contrast, UCB-based gossip algorithms need to prioritize under-sampled arms, requiring complex detection mechanisms and making them harder to implement and analyze under heterogeneous feedback. Our approach provides a simpler and more robust solution tailored for this setting.
>
> (ii) Refined weight matrix: The design of the weight matrix $W_t$ is another key aspect of our algorithm’s novelty. As defined in Eq.(2), we construct $W_t$ using the Laplacian matrix, which ensures that the weight matrices are independent and identically distributed (i.i.d.) across rounds. This property is essential for satisfying the conditions in Lemma B.3, enabling us to rigorously show that information from different agents becomes sufficiently balanced over time. As a result, the local estimates $z_{i,k}(t)$ on each agent converge to the true global mean reward $\mu_k$ as the number of rounds increases.
>
> (iii) Confidence interval design: Furthermore, we propose a *novel confidence interval* (Eq.~(4)) that captures both *estimation error* and *consensus error*, which is critical under the stochastic communication graph and heterogeneous reward model. Specifically, the estimation error term $\sqrt\frac{4\log(T)}{N\max(T\_{i,k}(t)-KL^\*,1)}$
> accounts for the statistical variance due to finite sampling, while the consensus error term $\frac{4(\sqrt{N}+\tau^*)}{\max(T_{i,k}(t)-KL^\*,1)}$ reflects the additional approximation error incurred due to time-delayed information propagation through the gossip protocol under a random graph. To our knowledge, this is the first formulation of a confidence bound that explicitly captures this two-fold structure in the heterogeneous MA-MAB setting.
>
> **Analytically, we introduce the following contribution.**
>
> Moreover, the analysis and theoretical results of our algorithm form a central—and arguably the most crucial—part of our contribution. In Theorem 5.3, we establish an upper bound on the global regret for the heterogeneous MA-MAB problem under random graph communication. The first term in the bound, $O\left(\sum_{k:\Delta_k>0}\frac{\log T}{\Delta_k} \right)$, reflects the estimation error originating from limited samples and matches the theoretical lower bound in Theorem 5.9, confirming its optimality. The second and third terms, $O\left( \frac{N^2\log T}{p\lambda_{N-1}(\text{Lap}(\mathcal{G}))}+\frac{KN^2\log(NT)}{p} \right)$, stem from consensus errors and reveal the regret’s dependence on key graph parameters, including connection probability $p$ and algebraic connectivity $\lambda_{N-1}(\text{Lap}(\mathcal{G}))$. As the first work to consider MA-MAB with general random graphs where each edge appears independently with arbitrary probability $p \in (0,1]$, we also provide the first regret bounds that explicitly reflect these structural dependencies. Remark 5.6 highlights the inherent trade-off between communication frequency and regret. As previously discussed, this trade-off is fundamental: for example, in a cycle graph with edge appearance probability $p$, information may require $O(N/p)$ rounds to propagate (similar philosophy as suggested by the reward delay in [6]). Since each agent must access global information to achieve optimality, this implies a fundamental regret lower bound of $O(N^2/p)$. Finally, as shown in Remark 5.5, our results generalize existing works as special cases (e.g., fixed or strongly connected graphs), underscoring the broad applicability of our framework.
>
> Together, these innovations enable us to systematically address both reward heterogeneity and random graph communication—an unexplored combination in prior work.
>
> [1] Mengfan Xu, and Diego Klabjan.
> Decentralized randomly distributed multi-agent multi-armed bandit with heterogeneous rewards. Advances in Neural Information Processing Systems, 2023.
>
>
> [2] Zhaowei Zhu, Jingxuan Zhu, Ji Liu, and Yang Liu. Federated bandit: A gossiping approach. Proceedings of the ACM on Measurement and Analysis of Computing Systems, 2021.
>
> [3] Jingxuan Zhu, and Ji Liu. Distributed multiarmed bandits. IEEE Transactions on Automatic Control, 2023.
>
> [4] Mengfan Xu, and Diego Klabjan. Multi-agent multi-armed bandit regret complexity and optimality. The 28th International Conference on Artificial Intelligence and Statistics, 2025.
>
> [5] David Martínez-Rubio, Varun Kanade, and Patrick Rebeschini. Decentralized cooperative stochastic bandits. Advances in Neural Information Processing Systems, 2019.
>
> [6] Tal Lancewicki, Shahar Segal, Tomer Koren, Yishay Mansour. Stochastic multi-armed bandits with unrestricted delay distributions. In International Conference on Machine Learning, 2021.

---

> > ### Comment · Reviewer_nBgW · 2025-08-04
> >
> > Thank you for your answers.
> >
> > I recommend that the authors incorporate an extensive (and technically detailed) discussion about the actual existence of the trade-off in the regret bounds. As I agree that the paper presents some technical contributions, I decided to raise my score accordingly.

---

> > > ### Author Response · Authors · 2025-08-04
> > >
> > > We sincerely thank the reviewer for carefully reading our response, for the valuable suggestions, and for reconsidering the score. We will incorporate the suggestions into the revised version. Specifically, we will 1) move the future work section to the main body, 2) add a technical discussion on the regret-communication trade-off using the above elaborations, 3) highlight the methodological contributions in a separate subsection (in the methodology section), and 4) include the above discussion on the tightness of the result in the analysis section as a remark (after the lower bound proposed in Section 5.2).

---

### Official Review · Reviewer_bG2e · 2025-07-02

**Clarity:** 3
**Significance:** 3
**Originality:** 2
**Rating:** 4
**Confidence:** 4

**Summary:**

This paper studies the heterogeneous multi-agent bandit setting, where the goal of each bandit is to minimize regret with respect to the globally best arm. The agents are assumed to exist on a fixed graph $\mathcal{G}$, and communicate with neighbors with probability $p$.

The authors propose an arm elimination algorithm that uses gossip to reach a consensus. The paper provides an upper bound on the regret of the algorithm, $\mathcal{O}\left(\sum_{k:\Delta_k>0}\Delta_k^{-1}\log(T)+p^{-1}\lambda^{-1}N^2\log(T)+p^{-1}N^2\log(T)\right)$, where $\lambda$ is the spectral graph of the graph $\mathcal{G}$, that neatly decomposes into a centralized regret bound and a consensus/decentralization term.

The authors provide a lower bound for the centralized regret that matches their upper bound. Lastly, experiments to demonstrate the algorithm are conducted.

**Questions:**

1. How tight is the consensus/gossip upper bound? Would it be possible to come up with a lower bound that includes the communication cost for some specific graph, like a tree or a cycle?

2. In Lemma B.4, is the $K$ in the upper bound necessary? The "waves" for each of the arms should happen in "parallel", which should render the argument in lines 512 to 515 unnecessary.

3. How is the second equality in equation (14) (line 531-532, the equality after the definition) obtained?

**Ethical Concerns:**

["NO or VERY MINOR ethics concerns only"]

**Final Justification:**

The authors have addressed my questions. Thus I am increasing my score.

**Limitations:**

Yes

**Quality:**

2

**Strengths And Weaknesses:**

Strengths:

1. The authors provide a stronger regret bound compared to many papers analyzing similar settings. Previous works have obtained regret bounds such as $\mathcal{O}\left(\sum_{k:\Delta_k>0} \Delta_k^{-1}N\log(T)\right)$ in [1]. The algorithm presented in this paper has a superior bound of $\mathcal{O}\left(\sum_{k:\Delta_k>0} \Delta_k^{-1}\log(T) + KN\log(T)\right)$.

2. The algorithm is simpler than many similar algorithms and uses fewer hyperparameters. Previous algorithms require a burn-in time $L$, with $L$ being a complicated function of other hyperparameters. These algorithms also require $p>1/2$. Furthermore, the gossip mechanism is simplified with each bandit maintaining a local record of the values of the arms, which is communicated over the graph.

3. Previous works do not capture the consensus/communication term of the regret as neatly as the regret bound presented. In [1], the communication cost is largely captured by the burn-in period $L$, which does not depend directly on the graph structure.

[1] Xu, Mengfan, and Diego Klabjan. "Decentralized randomly distributed multi-agent multi-armed bandit with heterogeneous rewards." Advances in Neural Information Processing Systems 36 (2023): 74799-74855.

Weaknesses:

The motivation for seeking global optimality rather than local is lacking. It would be best to include a statement, i.e., that the local bandits might have biases, to clarify this.

The experiments are lacking. In particular:

1. There should be more experiments with varied time horizons $T$, with comparisons to the upper bound. Specifically, Figure $1$ records the running regret given a fixed horizon $T$. Another graph showing the terminal regret for different horizons $T$ may better capture the dependence of the regret on $T$.

2. Figure 2 would benefit from more data points, particularly with $p$ close to $0$. Figure $2$ may also benefit from a log-scale, and linear fits to verify the $p^{-1}$ relation of the regret.

3. The experiments do not sufficiently capture the dependence on the spectral gaps of the base graph. A graph of the regret v/s the spectral gaps for a family of graphs, e.g., expander graphs, might be useful to verify the dependence on the graph structure.

Further, the conclusion and future work section of the paper has been included in the appendix rather than the main body.

---

> ### Author Rebuttal · Authors · 2025-07-31
>
> Thank you for your feedback and hope our responses address your concerns. If this is the case, we would appreciate it if you can reconsider your score.
>
> **Q1:** Position of the conclusion and future work
>
> **A1:** Thank you for pointing that out. We will reorganize the paper and move the conclusion and future work section to the main body as the final section in the revised version.
>
> **Q2:** Motivation of global optimality.
>
> **A2:** Thank you for this suggestion. In many real‐world applications, global rather than local optimality is the true objective. For instance, a company with multiple branches may face region‐specific demands: a policy that maximizes one branch’s profit could inadvertently harm others. To avoid such "local bias", each branch must choose a policy that maximizes the **aggregate** reward across all branches.
>
> Motivated by the real-world applications, we model this problem as $N$ agents seeking to maximise the **global reward**, as the sum of each agent’s expected reward. If each agent only optimizes its own local bandit, these local biases can cause the global regret to grow linearly in $T$ (as illustrated in [1], where linear regret is shown to be unavoidable if the graph is disconnected, preventing optimization toward global optimality). By contrast, our GSE algorithm is designed to minimise this global regret. As established in Lemma 5.2 and Theorem 5.3, both the per‐agent and total global regret under GSE converge at a logarithmic rate, demonstrating that GSE indeed achieves true global optimality rather than merely local gains.
>
> **Q3:** Additional experiments
>
> **A3:** Based on the suggestions, we conduct additional experiments as follows. For each experiment, we run the algorithm for 20 times and report the average performance.
>
> (i) To further investigate the regret’s dependence on the time horizon $T$, we report in Table 1 the final cumulative regret of GSE and DrFed-UCB across a range of $T$ values. The results show that GSE scales much more slowly with $T$ compared to DrFed-UCB, aligning with our theoretical tighter upper bound. This experiment confirms the favorable scaling behavior of GSE and supports the tightness of our regret analysis.
>
> *Table 1: Regret of GSE vs. DrFed-UCB for varying T with classical ER graph and p=0.9.*
> |Algorithms\T|1000|2000|3000|4000|5000|6000|7000|8000|9000|10000|11000|12000|
> |------------|----:|----:|----:|----:|----:|----:|----:|----:|----:|-----:|-----:|-----:|
> |**GSE**|83.64|91.53|97.03|100.20|103.35|104.94|106.81|108.03|110.21|111.42|113.01|113.00|
> |**DrFed-UCB**|158.04|243.58|305.32|354.99|393.03|425.75|453.93|481.16|499.52|520.04|527.65|529.93|
>
>
> (ii) To better understand the dependence of the regret on the link probability \(p\), we conduct additional experiments on complete base graphs with small values of $p\in[0.04, 0.18]$, and report the log–log relationship between $p$ and the resulting regret. The results are shown in the Table 2. By performing a linear regression between $\log(p)$ and $\log(\text{Regret})$, we obtain a slope $\hat\alpha = -0.93$ with $R^2 = 1.0$, indicating a nearly perfect linear fit (of order $\frac{1}{p^{0.93}}$). Surprisingly, we also report the $R^2$ corresponding to the curve of fit $\frac{1}{p}$, and find out that $R^2$ is $0.995$, which represents that the curve fit is quite statistically significant and thus a perfect linear fit. This strongly supports the inverse proportionality between regret and $p$, i.e., regret scales approximately as $O(1/p)$, which is consistent with our theoretical upper bound. This result empirically also validates that smaller $p$ leads to significantly higher regret due to slower information diffusion across agents.
>
> *Table 2: Log–log data for GSE: $\log(p)$ and $\log(\mathrm{Regret})$ with classical ER graph, p=0.9)*
> |$p$|0.04|0.06|0.08|0.10|0.12|0.14|0.16|0.18|
> |---|----|----|----|----|----|----|----|----|
> |**$\log(\mathrm{Regret})$**|7.4019|7.0136|6.7405|6.5331|6.3653|6.2232|6.1045|6.0000|
> |**$\log(p)$**|–3.2189|–2.8134|–2.5257|–2.3026|–2.1203|–1.9661|–1.8326|–1.7148|
>
> **Fit allowing slope free**
> |Statistic|Value|
> |---------|-----|
> |Slope $\hat\alpha$|–0.93|
> |Intercept $b$|4.39|
> |$R^2$|1.00|
>
> **Fit with fixed slope = –1**
> |Statistic|Value|
> |---------|-----|
> |Slope $\hat\alpha$ (fixed)|–1.00|
> |Intercept $b$|4.236|
> |$R^2$|0.995|
>
>
> (iii) To empirically investigate the dependence of regret on the spectral gap of the base graph, we conduct experiments using $d$-regular graphs with varying degrees $d$. Specifically, we construct the base graphs as circulant graphs. This structure provides a controllable family of regular graphs with increasing algebraic connectivity as $d$ grows. We run our algorithm GSE under this setting with the link probability $p = 0.9$. Table 3 shows a clear inverse relationship: as $\lambda_{N-1}$ increases with higher $d$, the regret decreases significantly. This supports our theoretical findings that larger algebra connectivity leads to more efficient information propagation and thus lower regret.
>
> *Table 3: Average regret of GSE with d-regular graph*
>
> |d-regular graph|2|4|6|8|10|12|14|
> |---------------|--|--|--|--|--|--|--|
> |$\lambda_{N-1}$|0.1716|1.39|1.97|3.97|6.73|10.15|14.00|
> |$\mathrm{Regret}$|751.28|255.52|165.66|133.92|121.93|115.12|112.47|
>
> **Q4:** Regarding the tightness of bounds and possibility to include communication cost of graph in lower bound
>
> **A4:** Thank you for the insightful question. In our paper, we provide a theoretical regret upper bound in Theorem 5.3 and a corresponding lower bound in Theorem 5.9. The first term in the upper bound, which captures the centralized regret due to the statistical uncertainty from limited samples, matches the lower bound, indicating that this term is tight and optimal. The remaining two terms in the upper bound arise from delays in information propagation due to random communication, and these are not yet matched by a lower bound.
>
> To gain intuition on the potential tightness of these communication-related terms, consider a simple cycle graph where each edge is activated independently with probability $p$ at each round. On average, it would take $O(N/p)$ rounds for information from one agent to reach all the other agents (can be viewed as reward delay in [3]). Since agents are optimizing for global rewards, one agent must wait to receive information from all others to make informed decisions. This implies that each agent may need $O(N/p)$ rounds to effectively optimize its local policy with respect to the global objective. Summing across all $N$ agents gives a global regret contribution of at least $O(N^2/p)$, which intuitively explains the dependency on $p$ and $N$ in the graph-dependent terms in our upper bound. This argument provides heuristic evidence for partial tightness of our bound with respect to communication regret.
>
> We would also like to add that we make contributions toward narrowing the gap between the lower and upper bounds in existing work. Notably, our upper bound is smaller compared to [2]. More importantly, it also introduces a novel reflection of the communication graph through the presence of $\frac{1}{p}$ in the upper bound. Empirically, Table 2 in A3 shows that the actual regret exhibits a dependency of $\frac{1}{p}$, which numerically validates the near-tightness of our upper bound. On the other hand, our lower bound improves upon existing work [1] by explicitly incorporating the sub-optimality gap. We acknowledge that a formal lower bound capturing communication costs—especially for specific topologies such as trees or cycles—remains open and is an important direction for future work.
>
> **Q5:** Regarding the necessity of $K$ in Lemma B.4
>
> **A5:**
> We thank the reviewer for this insightful comment. Indeed, under our round‐robin protocol, the “waves” for different arms proceed **in parallel**, so we can refine the disagreement‐length bound as follows.  At any time $t$, if $|\mathcal S_i(t)|=m$, each wave lasts at most $\left\lceil \frac{N L_p(\delta)}{m} \right\rceil$ rounds for agent $i$. Since $|\mathcal S_i(t)|$ decreases by at least one each time a wave completes, the total number of disagreement rounds is bounded by
> $\sum_{m=1}^K  \left\lceil \frac{N L_p(\delta)}{m} \right\rceil=NL_p(\delta)\sum_{m=1}^K \frac{1}{m} + O(K)
> =NL_p(\delta)(H_K + O(1)),
> $
> where $H_K=\sum_{m=1}^K 1/m=O(\log K)$.  Consequently, the right‐hand side of Lemma B.4 can be improved toO(NL_p(\delta)\log K).$ We appreciate the suggestion and will incorporate this tighter bound and its detailed proof in the revised version of the paper.
>
> **Q6:** The derivation of Eq.(14) (line 531-532)
>
> **A6:** The second equality holds because of the definition of $z_{i, k}(t)$ and the doubly stochasticity of the matrix of $W_t$.
>
> References
>
> [1] Mengfan Xu, and Diego Klabjan. Multi-agent multi-armed bandit regret complexity and optimality.
> In International Conference on Artificial Intelligence and Statistics, 2025.
>
> [2] Mengfan Xu, and Diego Klabjan.
> Decentralized randomly distributed multi-agent multi-armed bandit with heterogeneous rewards. Advances in Neural Information Processing Systems, 2023.
>
> [3] Tal Lancewicki, Shahar Segal, Tomer Koren, Yishay Mansour. Stochastic multi-armed bandits with unrestricted delay distributions. In International Conference on Machine Learning, 2021.

---

> > ### Comment · Reviewer_bG2e · 2025-08-01
> >
> > Thanks for the detailed response above. I have another question re. the heterogeneous setting.
> >
> > Perhaps a better term for this setting would be a noisy homogeneous setting? In my mind, a heterogeneous setting is one where groups of agents have different best arms, and the problem is to form a consensus among the agents with the same best arm. I realize this is not in the scope of this paper, and that is fine. My only question is re. the specific use of the term 'heterogenous setting'. Can the authors please comment.

---

> > > ### Author Response · Authors · 2025-08-02
> > >
> > > Thank you so much for reading the rebuttal and for the great question. We would like to add that our setting is exactly as the reviewer has described, namely the “heterogeneous setting”: local agents may have different local optimal arms based on their local reward means $ \mu_{i,k} $ (where $ i $ denotes agent $ i $), which can differ from the global optimal arm determined by the global reward mean $ \mu_k = \frac{1}{N}\sum_{i=1}^N \mu_{i,k}$,  the average of all agents’ local reward means. Thus, agents must communicate to infer the global reward mean and reach consensus on the global optimal arm.
> > >
> > > For illustration purposes, we include a concrete example with 4 agents and 5 arms (as shown in the following table). Each cell shows the local reward mean of an arm for a given agent—rows correspond to agents, and columns to arms. The cells of the local optimal arms for agents 1, 2, 3, and 4 are highlighted in bold, i.e., arms 1, 2, 3, and 4, respectively. Specifically, agent 1's local optimal arm is arm 1 (0.90), agent 2's local optimal arm is arm 2 (0.90), agent 3's local optimal arm is arm 3 (0.90), and agent 4's local optimal arm is arm 4 (0.90). However, based on the global reward means (the column-wise averages in the last row), arm 5 is globally optimal, also in bold. In other words, the highest global reward mean is on arm 5 (0.40), even though no individual agent identifies it as its local optimal arm. This implies that no agent can identify the global optimal arm without communicating and inferring the global reward means. While doing so, the randomness and potential disconnectivity in the communication graph $ G_t $, as well as the randomness and bias in the noisy reward observations $ X_{i,k}(t) $ (agents observe only $ X_{i,k}(t) $, not $ \mu_{i,k} $; $ \mu_{i,k}$ is biased compared to $ \mu_k$), further complicate this task.
> > >
> > > | Agent \ Arm | Arm 1 | Arm 2 | Arm 3 | Arm 4 | Arm 5 |
> > > |------------|-------|-------|-------|-------|-------|
> > > | Agent 1    | **0.90** | 0.10 | 0.10 | 0.10 | 0.40 |
> > > | Agent 2    | 0.20 | **0.90** | 0.20 | 0.20 | 0.40 |
> > > | Agent 3    | 0.10 | 0.20 | **0.90** | 0.20 | 0.40 |
> > > | Agent 4    | 0.10 | 0.10 | 0.20 | **0.90** | 0.40 |
> > > | **Global** | 0.33 | 0.33 | 0.35 | 0.35 | **0.40** |
> > >
> > > If we understand it correctly:
> > > * Noisy homogeneous: agents' local rewards may have different mean values, but the local optimal arms are the same, i.e., order-preserving with respect to arms; thus, the global optimal arm coincides with the local optimal arm.
> > > * Fully homogeneous: agents' local rewards have identical mean values, which naturally leads to the same local optimal arm which is also globally optimal.
> > >
> > > The necessity of communication among agents follows the order: heterogeneous > noisy homogeneous > fully homogeneous.
> > >
> > > We would appreciate your clarification if our understanding of the noisy homogeneous setting differs from what the reviewer intended, and we are happy to answer any further questions the reviewer may have.
> > >
> > > Thank you once again for your feedback, and we hope our responses address your concerns. If so, we would greatly appreciate it if you could reconsider your score.

---

> > > > ### Comment · Reviewer_bG2e · 2025-08-05
> > > >
> > > > By heterogeneous, I meant that different agents have different local optimal, and that there is no single global optimal arm. So the bandit problem would decompose into many global sub problems.
> > > >
> > > > The noisy homogeneous setting (for me) is one where different agents have different local optimal, and that there is single global optimal arm that has highest average reward across arms.
> > > >
> > > > (Note this is just a language question, not a technical question). In that sense, place your work in the noisy homogeneous setting.
> > > >
> > > > In any case, I am happy with the rebuttal and will  increase my score.

---

> > > > > ### Author Response · Authors · 2025-08-06
> > > > >
> > > > > Thank you so much for reading the responses, for the detailed suggestions and clarifications, and for reconsidering the score. The discussions have been valuable and helpful. Yes, based on the reviewer’s set of definitions, our work belongs to the “noisy homogeneous setting.” We believe the suggested heterogeneous setting is also a valuable direction for future work, given the mixed motives/objectives of real‑world decision makers (which might also involve equilibrium considerations from game theory). We will include this sentence in the future work section as well.
> > > > > Also, based on the discussion, we will 1) include the above discussion on the tightness of the result and communication in the analysis section as a remark (after the lower bound proposed in Section 5.2), 2) include the above additional experiments, 3) include the analysis regarding $K$ in Lemma B.4, and 4) move the conclusion and future work section to the main body.
> > > > > Once again, we truly appreciate the thoughtful suggestions.

---

### Decision · Program_Chairs · 2025-09-17

**Decision:**

Accept (poster)

**Comment:**

The paper studies the multi-agent multi-armed bandits problem over random communication graphs (Erdos-Renyi, each edge appears with probability p). The authors propose an algorithm that achieves roughly log T regret and has dependence on the connectivity of the communication graph, the number of agents and arms and provide a matching lower bound. The reviewers were engaged in the discussion and provided thorough comments. The overall opinion was positive after the rebuttal. We recommend acceptance and advise the authors to invoke the reviewers' thorough comments in the camera ready version.